# Hybrid whale algorithm with evolutionary strategies and filtering for high-dimensional optimization: Application to microarray cancer data

**Rahila Hafiz**👤 *, Sana Saeed

College of Statistical Sciences, University of the Punjab, Lahore, Pakistan

* rahila_ahsan84@yahoo.com

## Abstract

The standard whale algorithm is prone to suboptimal results and inefficiencies in high-dimensional search spaces. Therefore, examining the whale optimization algorithm components is critical. The computer-generated initial populations often exhibit an uneven distribution in the solution space, leading to low diversity. We propose a fusion of this algorithm with a discrete recombinant evolutionary strategy to enhance initialization diversity. We conduct simulation experiments and compare the proposed algorithm with the original WOA on thirteen benchmark test functions. Simulation experiments on unimodal or multimodal benchmarks verified the better performance of the proposed RESHWOA, such as accuracy, minimum mean, and low standard deviation rate. Furthermore, we performed two data reduction techniques, Bhattacharya distance and signal-to-noise ratio. Support Vector Machine (SVM) excels in dealing with high-dimensional datasets and numerical features. When users optimize the parameters, they can significantly improve the SVM's performance, even though it already works well with its default settings. We applied RESHWOA and WOA methods on six microarray cancer datasets to optimize the SVM parameters. The exhaustive examination and detailed results demonstrate that the new structure has addressed WOA's main shortcomings. We conclude that the proposed RESHWOA performed significantly better than the WOA.

## 1. Introduction

Metaheuristics are a set of tactics used to navigate the search space. These tactics are the natural processes becoming more critical in genetic engineering [1]. The primary goal is rapidly searching the search space for near-optimal solutions to a given problem. Nature-inspired metaheuristics include evolutionary, physics-based, and swarm-based algorithms that effectively solve complex optimization problems [2]. **Evolutionary strategy-based** algorithms mimic genetic behaviour and generate innovative solutions iteratively through mutation and recombination. They select the best individuals from the population and carry them over to

Ovarian, and Leukemia were downloaded from https://csse.szu.edu.cn/staff/zhuzx/datasets.html, while the Carcinoma tumour data set is come from [ Princeton University gene expression project] at http://genomics-pubs.princeton.edu/oncology.

**Funding:** The authors received no specific funding for this work.

**Competing interests:** The authors have declared that no competing interests exist.

the next generation, repeating this process until they obtain a satisfactory result. The two types of ES are non-recombinative and recombinative. In evolutionary strategy, non-recombinative strategies involve mutation-only operators that modify the parent solution to produce a new solution. In contrast, recombinative methods involve recombination operators that combine the features of multiple parent solutions to create a new offspring solution. Genetic algorithms (GA) [3] and Genetic programming (GP) [4] are the most common evolutionary-based approaches. **Physics-based** algorithms simulate physical occurrences in space. Simulating annealing (SA) is a popular method in this field [5]. **Swarm-based** algorithms are nature-inspired optimization algorithms that mimic the collective behaviour of social organisms such as ants, bees, birds, and fish. Particle swarm optimization (PSO) is a well-known swarm-based method that mimics swarm social behaviour. The field began with the expansion of GA, and since then, researchers have simulated numerous versions. In the literature, several optimization algorithms have been proposed [6]. Due to their ability to achieve the optimal global solution with fewer parameters, these algorithms are widely utilized across various fields [3, 7–9].

Mirjalili, an Australian researcher, introduced a novel heuristic population-based algorithm in 2016 that mimics whale hunting behaviour, distinguished by a distinctive spiral pattern known as bubble net feeding. This feature gives this algorithm a significant advantage over others [2]. It uses probability to update its optimal individual and motion modes, resulting in greater randomness, faster convergence speed, and a sound effect in practical engineering. It is currently widely used in images [7, 10], medical [8, 9], microgrids [11], and other fields. However, it has the same drawbacks as different swarm intelligence optimization algorithms. When dealing with complex environmental problems, it is easy to become bogged down by local optimum, low convergence rate, and low precision. Given the limitations of WOA, scholars and experts have devoted their efforts to improving the algorithm. To further balance the development and exploration stages in the traditional WOA algorithm, Sahu et al. proposed the MWOA algorithm. They use MWOA as a static synchronous series compensator of the multi-input-single-output (MISO) type (SSSC) [12]. By incorporating a chaos strategy, Sayed et al. proposed a chaotic whale optimization algorithm (CWOA) that improved the ability to jump out from the optimal local solution [13]. Yan et al. proposed a method (AWOA) for using logistic mapping to initialize population positioning and inertia weight to improve population diversity and accelerate convergence speed [14]. Several current WOA improvement algorithms have improved the optimization effect compared to the traditional WOA algorithm, but WOA performance still has much room for enhancement. Researchers need to conduct more research on balancing the ability of local and global exploration, quickly falling into the local optimum, improving convergence accuracy, and so on.

Among the three major branches of evolutionary computation, genetic algorithms (GAs), evolutionary programming (EP), and evolution strategies (ESs), ESs are the only one that was initially proposed for numerical optimization and is still widely used in optimization today [15, 16]. ESs primarily use mutation as the search operator, although they have also used recombination. The state-of-the-art of ES is $(\mu, \lambda)-$ES, where $\lambda > \mu \geq 1$, $(\mu, \lambda)$ means that $\mu$ parents generate $\lambda$ offspring through recombination and mutation in each generation. The best $\mu$ offspring are selected deterministically from the $\lambda$ offspring and replace the parents [17]. The strategies do not use elitism and probabilistic selection. This paper only considers a simplified version of ESs, i.e., RES, without mutation and elitism. ESs are population-based versions of generate-and-test algorithms [18]. They generate fresh solutions using search operators such as mutation and then use a selection scheme to determine which newly developed solutions should survive for the next generation. The advantage of viewing ESs as a variant of search algorithms is that the relationships between different search algorithms, such as simulated annealing (SA), tabu search (TS), hill-climbing, etc., we can make it more explicit

and thus easier to explore. Furthermore, the generate-and-test perspective on EAs clarifies that genetic operators like a crossover (recombination) and mutation are stochastic search operators used to generate new search points in a search space. Rather than biological analogy, its ability to produce promising new facts with a higher probability of leading to a global optimum best describes a search operator's effectiveness.

The function of a test in a generate-and-test algorithm or selection in an EA is to determine how promising a new point is. These assessments can be heuristic or probabilistic. The $(\mu, \lambda)$ −ESs use a Gaussian mutation to generate new offspring and the deterministic selection to evaluate them. There has been a lot of work on different selection schemes for ESs [17].

The creation of prediction models is a fascinating application of machine learning. Prediction models have been used in several biological applications [19–22]. Using gene expression profiles to identify and classify malignant and normal tissues can be a challenging application of machine learning. Still, its difficulty may vary depending on the specific context and data type being analyzed [23]. The novel DNA microarray technique can detect the expression levels of several genes in a single experiment. Researchers can use this technology to understand the genes expressed in each tissue under various conditions. The Support Vector Machine (SVM) is widely used in machine learning models [24] and is a supervised learning algorithm for classification and regression analysis. It performed well in various classification applications [25–29]. Medical diagnosis is an essential application for the SVM classifier because it is crucial in diagnosing specific disorders. We must first solve the SVM model to benefit from it, including determining the best parameters. Many solutions have been developed in recent years to address this difficulty, such as probabilistic optimization methods, Laplace evidence approximations, and Generalized Approximate Cross-Validation (GACV) error [30, 31].

Accurate tumour progression prediction is critical for cancer diagnosis and treatment. Developing cDNA microarray technology is significant in molecular biology and cancer research [32]. Because of the enormous number of genes monitored, cDNA microarray data sets have a high dimensionality, and there are frequently few samples. To improve the efficiency of the considered model SVM, we employ the Bhattacharyya distance (BC) and signal-to-noise ratio (SNR) statistical filtration techniques [33].

There can be a strong correlation between high-dimensional data, such as microarray data, and optimization techniques. High-dimensional data typically involve datasets with many features or variables, posing several challenges, including noise, redundancy, and the curse of dimensionality. Optimization techniques can be valuable in addressing these challenges and extracting meaningful information from such data.

Previous studies have used a few ways in which optimization is often applied to high-dimensional data, especially in microarray analysis [34–37]. High-dimensional data often contain many irrelevant or redundant features. **Feature selection** techniques use optimization algorithms to identify a subset of the most informative features, reducing dimensionality while preserving the relevant information [38–44]. **Dimensionality reduction** methods like Principal Component Analysis (PCA) and t-distributed Stochastic Neighbor Embedding (t-SNE) aim to project high-dimensional data into a lower-dimensional space while preserving the most significant variance or structure [37, 45–48]. These techniques often involve optimization to find the best projections. In microarray data analysis, **clustering and classification** tasks are common. Machine learning practitioners employ optimization to find optimal parameters for machine learning algorithms like k-means clustering, support vector machines (SVM), or neural networks, which can handle high-dimensional data for tasks such as identifying disease subtypes or predicting outcomes [49, 50]. When building predictive models from high-dimensional data, **regularization and model selection** techniques like L1 regularization (Lasso) and L2 regularization (Ridge) use optimization to balance model complexity and accuracy [51–53].

Model selection methods also rely on optimization to choose the best model hyperparameters. **Gene Network Inference:** Researchers often use microarray data in genomics to infer gene regulatory networks. Optimization techniques can help discover the relationships between genes by fitting models that best explain the observed expression patterns [54–56]. **Biomarker Discovery:** High-dimensional data analysis is crucial in identifying biomarkers for disease diagnosis, prognosis, or treatment response [57–59]. Optimization plays a role in feature selection and model building for biomarker discovery. **Optimization in Experimental Design:** When planning microarray experiments, optimization techniques can assist in selecting the most informative samples or conditions to maximize the utility of the data collected [60, 61].

The correlation between high-dimensional data, such as microarray data, and optimization techniques is significant. Optimization methods are essential for the preprocessing, analyzing, and modelling high-dimensional datasets to extract meaningful information, reduce noise, and enhance the effectiveness of data-driven decision-making processes.

This paper presents a new hybrid algorithm, the Recombinative Evolutionary Strategy Hybrid Whale Optimization Algorithm (RESHWOA), to tackle the problems mentioned earlier. The following are the paper's main contributions:

1. The authors present a new optimization algorithm incorporating a recombinant evolutionary strategy into the whale optimization algorithm to improve the diversity of the positions of the "whales," ideally leading to better optima. This approach can significantly reduce the occurrence of getting stuck in local optima and improve the convergence accuracy of the algorithm.

2. The author's main contribution is comparing the proposed algorithm with the original version of WOA to validate its effectiveness on benchmark functions. Additionally, the authors conduct scalability experiments and present the results in Tables 3 and 4, demonstrating the algorithm's ability to solve high-dimensional problems.

3. Inspired by Table 6 algorithms and techniques, this research primarily aims to optimize the hyperparameters of the SVM model by utilizing the proposed method and the original WOA optimizer to minimize the MSE. The main objective is to identify the optimal set of hyperparameters for high-dimensional data with minimum MSE.

4. We ran this algorithm on various microarray cancer datasets and examined the results with thirty runs to check the efficiency of the considered model.

5. To reduce the dimensionality of the data, we have employed statistical filtration techniques such as Bhattacharyya distance (BC) and signal-to-noise ratio (SNR) [33, 62] in our approach.

The paper comprises five sections. In Section 2, we illustrate the main ideas of the standard whale algorithm and standard recombinative evolutionary strategies, and then we describe the details of our proposed hybrid algorithm in Section 3. In Section 4, we will demonstrate and analyze the experimental results. Finally, Section 5 concludes the work.

## 2 Whale optimization and recombinative evolutionary strategy

The whale optimization algorithm (WOA) is a new nature-inspired metaheuristic optimization algorithm that Australian scholar Mirjalili and others proposed. The main inspiration of the algorithm is to simulate the predation behaviour of the humpback whale population and update the position of the candidate solution through the process of the whale population, spiral updating position, encircling, and finding prey. Recombinative evolutionary strategy

(RES), developed by Deb in 1997, is a metaheuristic technique [12, 30]. Both WOA and RES are non-gradient-based evolutionary algorithms that potentially have a parallel structure. Research has proven that both algorithms can get better optimization results than existing methods.

## 2.1 Whale optimization algorithm

The whale optimization algorithm is a relatively new nature-inspired optimization algorithm proposed in 2016 by Seyedali Mirjalili, Andrew Lewis, and Ashraf Alasty [2, 63]. Humpback whales′ hunting behaviour forms the basis for it, where whales cooperate to encircle their prey and gradually decrease the prey escape options until they catch it. In the WOA algorithm, researchers represent each potential solution as a whale, and the whale's position corresponds to the decision variables in the optimization problem. The algorithm begins with an initial population of randomly generated whales, and the goal is to find the best solution that minimizes or maximizes the objective function [10]. The WOA algorithm employs three mechanisms to explore and exploit the search space: exploration, exploitation, and convergence. During the exploration phase, the whales move randomly to explore the search space. In the exploitation phase, the whales swim and encircle to achieve the optimal solution. Finally, in the convergence phase, the whales use the spiral and bubble-net hunting behaviour to converge to the optimal solution. The WOA algorithm also uses a set of adaptive parameters updated during the optimization process to control the balance between exploration and exploitation. Experts utilize these parameters to adjust the step sizes of the whales and the search agent search range to ensure that the algorithm converges efficiently and accurately.

The WOA algorithm effectively solves various optimization problems, including benchmark functions, engineering design problems, and machine learning problems. However, like other optimization algorithms, the performance of the WOA algorithm depends on the problem characteristics and the selection of appropriate algorithm parameters. The WOA is relatively easy to understand and codes in applications. Since its inception, WOA has gained widespread popularity and has found application in various engineering fields. Below, we describe in detail the steps of the algorithm in section 3.

## 2.2 Recombinative evolutionary strategy

Evolutionary strategies (ESs) are a powerful class of search and optimization methods inspired by natural evolutionary mechanisms [22, 64]. Intuitively, they mimic evolutionary principles such as a population-based strategy, information inheritance, information variability through crossover/mutation, and individual choice based on fitness. There are various evolutionary strategy variants; however, we will concentrate on discrete recombination (DR), a type of ES. In the DR phase, researchers develop a starting population of μ individuals, and during a DR generation, $\rho$ parents produce λ descendants. The algorithm repeatedly selects parents during this process. They recombine the components, and then they mutate the resulting offspring. In the primary ES, two parents produce a descendant, respectively. Initially, two individuals $P_1$ and $P_2$, are stochastically determined as parents of the offspring. All individuals have the same selection probability of 1/μ.

The components of both parent vectors subsequently recombine to produce offspring. Empirical evidence has shown that using different recombination schemes for the decision variables provides an advantage. To determine the values of the decision variables (Index O), we stochastically select the value of one or the other parent indices ($P_1$ and $P_2$) with equal probability for each decision variable. We refer to this approach as discrete recombination. It strongly resembles the Uniform Crossover in GA. On the other hand, intermediary

| Parent1 | 5 | 4 | 1.9 | 2 |

| Parent 2 | 7 | 3 | 3.6 | 5 |

| Offspring | 5 | 3 | 3.6 | 2 |

**Fig 1. Discrete recombination.**

recombination performs an averaging operation on the mutation step widths of parents in genetic algorithms to generate offspring [65]. Let us focus on one solution element in Eq 1: the $j^{th}$ element, denoted as $X_{O,j}$. This element can take on the value of either $X_{P_{1,j}}$ or $X_{P_{2,j}}$. Here, $P_1$ and $P_2$ are randomly selected parent indices from the population with μ individuals.

$$X_{O,j} = X_{P_{1,j}} \ or \ X_{P_{2,j}} \qquad (j = 1, 2, \ldots, \lambda \quad ; \quad p_1, p_2 \epsilon(1, 2, \ldots.\mu)) \qquad (1)$$

Let us consider a scenario with four parents ($P_1$, $P_2$, $P_3$, $P_4$) and $\lambda$ offspring. Each parent has values for several decision variables. For example, we derive a discretely recombined solution from the following four solutions.

$$P_1 : \ X_{P_{1,1}} X_{P_{1,2}} X_{P_{1,3}} X_{P_{1,4}} X_{P_{1,5}} \ldots \ldots \ldots X_{P_{1,\lambda}}$$

$$P_2 : \ X_{P_{2,1}} X_{P_{2,2}} X_{P_{2,3}} X_{P_{2,4}} X_{P_{2,5}} \ldots \ldots \ldots X_{P_{2,\lambda}}$$

$$P_3 : \ X_{P_{3,1}} X_{P_{3,2}} X_{P_{3,3}} X_{P_{3,4}} X_{P_{3,5}} \ldots \ldots \ldots X_{P_{3,\lambda}}$$

$$P_4 : \ X_{P_{4,1}} X_{P_{4,2}} X_{P_{4,3}} X_{P_{4,4}} X_{P_{4,5}} \ldots \ldots \ldots X_{P_{4,\lambda}}$$

$$\text{Offspring :} \qquad X_{P_{2,1}} X_{P_{4,2}} X_{P_{3,3}} X_{P_{1,4}} X_{P_{4,5}} \ldots \ldots \ldots X_{P_{4,\lambda}}$$

We randomly take the value from one of the parents to create an offspring for each decision variable. This process helps explore different combinations of decision variables to improve the quality of solutions in optimization problems potentially.

Figs 1 and 2 illustrate the recombination forms' working and further recombination variants. The practical implementation of the discrete recombination is also available in S1 Fig.

## 3. RESHWOA algorithm

The conventional WOA method generates the whales' initial solution using random numbers. We will apply the DR strategy, starting with an initial random population, to achieve a diverse perspective. Moreover, the computer-generated initial population is generally unevenly distributed in the solution space, resulting in a low initial population diversity [64]. It also necessitates a stochastic and iterative procedure to evolve a population of individuals over a predetermined number of generations [66]. In contrast, the proposed hybrid method practices a discrete recombinative technique and introduces a group of recombinants and sampling from the DR [6]. The main reason is to use the recombination principle; it is a popular method of combining the beneficial characteristics of the parents in the offspring and holds a

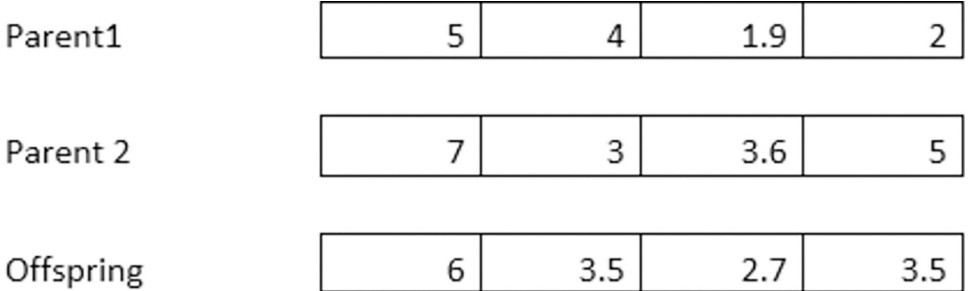

**Fig 2. Intermediate recombination.**

significant position in computing. Our proposed method provides a diverse initial population for improving the algorithm's global search capability and the accuracy of the optimal solution. The literature provides evidence for the effectiveness of hybrid optimization techniques for the initial population, which involve integrating multiple optimization algorithms to yield superior results. For example, in the CMA-ES method, new search points are generated by sampling from a multivariate normal distribution [67]. Similarly, in chaos initialization, a set of chaotic variables is generated as the initial population to improve the optimization performance of the WOA algorithm [68].

The structure of the WOA algorithm consists of the following components. Below, we describe three key aspects, each highlighted in bold.

**Step 1:** Generate the vectors of the initial WOA population. All the vectors are random values between 0 and 1; also, initialize k and i.

**Step 2: Encircling Prey Behaviour:** The location of the whale closest to the prey influences the

movement behaviour of other whales, making other whales' approach the optimal whale. The location update model is as follows.

$$D_1 = |C.X_{(*)}(i) - X(i)| \qquad (2)$$

$$X(i+1) = X_{(*)}(i) - A.D_1 \quad \text{if } p < 0.5 \qquad (3)$$

$$A = 2a.r_1 - a \qquad (4)$$

$$C = 2.r_1 \qquad (5)$$

Parameter 'a' accelerates from 2 to 0 and attains zero at the maximum iteration, $r_1$ randomly generated vector lies between 0 and 1. 'A' assumed the values -1 to 1 range. Where 'i' indicates the current iteration, A and C are coefficient vectors, X* is the position vector of the best solution obtained so far, and X is the position vector.

**Step 3: Attacking Prey Behaviour**: When attacking prey, the humpback whale has its unique path movement mode, which attacks targets by the bubble net movement. Its specific mathematical model is as follows:

$$X(i+1) = D'.e^{bl}.\cos(2\pi l) + X_{(*)}(i) \qquad \text{if } p \geq 0.5 \qquad (6)$$

$$D' = |X_{(*)}(i) - X(i)| \qquad (7)$$

Where b is the logarithmic helix shape constant, usually b is one, and l is the random number between [−1,1]. $D'$ indicates the distance of the ith whale to the prey.

It is worth mentioning that whether a whale population encircles its prey or spirals to attack prey depends on the p-value, and p is a random number between [0,1].

**Step 4: Searching Prey Behaviour**: To improve the global search ability of the optimization algorithm, whale′s populations surround prey when $|A|<1$ or randomly select a whale's position as a reference to update other whales′ positions when $|A|>1$. The ability is to move whale populations away from the position of the whale closest to prey so far for a global search. The specific mathematical model is as follows:

$$D_1 = |C.X_{(rand)}(i) - X(i)| \tag{8}$$

$$X(i+1) = X_{(rand)}(i) - A.D_1 \tag{9}$$

Where $X_{(rand)}(i)$ is a random whale position.

**Step 5:** If $p \geq 0.50$, update the position of the current search by Eq (6).

**Step 6:** If 'i' is not equal to no. of whales, then update 'i' and go to step 2. else

**Step 7:** if $k \neq$ no. of iterations, update k and go to step 2.

The genetic algorithm repeats the entire process until it achieves the desired number.

The proposed model actively enhances the initialization phase of WOA, thus leading to a more efficient optimization process. To develop the proposed model, let us delve into how we determine the values of decision variables for the offspring using discrete recombination.

**Step 1:** Initialize upper boundary (ub)

**Step 2:** Initialize lower boundary (lb)

**Step 3:** Initialize the dimensionality of the problem (dim).

**Step 4:** Calculate several boundaries (Boundary no) based on the size of ub.

**Step 5:** Set population size parameters $\lambda$ and $\mu$.

**Step 6:** Generate a random population (pop) of lambda individuals.

**Step 7:** Initialize an empty matrix 'pxx' with dimensions (lambda, lambda).

**Step 8:** Loop through each individual in the population.

**Step 8.1:** Eq 1 involves random sampling method to select $\mu$ individuals from 'pop' without replacement to form a group of $\rho$ parents.

**Step 8.2:** Stack these individuals into a matrix xm.

**Step 8.3:** Get dimensions of xm (nrow and ncol).

**Step 8.4:** Randomly select indices from each column of xm.

**Step 8.5:** Update the corresponding row in 'pxx' with selected values from 'xm.'

**Step 9:** Check if the boundaries of all variables are equal, and the user enters a single number for both ub and lb.

**Step 9.1:** Loop through each dimension. Scale each column of pxx by (ub—lb) and then add lb and get a matrix 'Positions'.

**Step 10:** If each variable has different lb and ub (Boundary no is greater than 1).

**Step 10.1:** Loop through each dimension. Scale each column of pxx by (ub—lb) and then add lb and get a matrix 'Positions'.

**Step 11:** Resulting matrix Positions contains the initial population of search agents.

**Step 12:** Return the Positions matrix as the initialized population.

## 3.1 Simulation experiments

We conduct a test setup to demonstrate how RESHWOA performs for both optimization evaluation metrics.

1. Unconstrained Unimodal Test Function

2. Unconstrained Multimodal Test Function

**3.1.1 Benchmark and experimental setup.**    Benchmark functions are an essential tool to evaluate the precision, convergence rate, robustness, and overall performance of new algorithms and their features. Therefore, we have selected a set of thirteen benchmark test functions based on their characteristics, modality, and other properties, providing various functions with varied difficulties. The benchmark functions used in our study are the same as those used in previous research [69, 70] and are summarized in Tables 1 and 2. The dimension of the benchmark function is denoted by D, the scales of the variables by S, and $F_{min}$ represents the global optimum value in the variable scales.

Before beginning the simulation study, we must determine four parameters for RESHWOA. The parent population size is one of the parameters, and the others are A, C, and D. Configure; these parameters are as follows: $\mu = 100$, $\lambda$ 100, $D = 30,100$, $\rho = 5$, and A and C are internally adjusted. We evaluated the functions over 30 runs using 500 iterations and 30 random search agents.

**3.1.2 Intensification capability experiment.**    Unimodal benchmark functions have a single global optimum over their entire domain. Experiments on unimodal benchmark functions revealed an intensification of convergence, as shown in Table 3. Results in Table 3 showed that the proposed RESHWOA algorithm would perform best in most cases. However, the WOA algorithm sometimes finds the global optima with the same iteration number. The best values are in bold.

**3.1.3 Diversification capability experiment.**    Multimodal benchmark functions contain multiple local optimal and one global optimum. When approaching these functions with limited exploration, local optima can easily trap individuals, insufficient search strategies, or premature convergence. To escape entrapment, individuals should possess diversification capability. Table 4 shows the results of simulation experiments that produce multimodal benchmark functions.

**3.1.4 Scalability experiment.**    Tables 3 and 4 compare the results of both dimensions for all the test functions considered in this article. We conducted a comparative study to justify the efficiency of the proposed method. We validate the performance of RESHWOA by analyzing the results in two parts. In the first part, the results with the 30-dimensions case show that RESHWOA produced the best values equal to the globally best values for two test functions ($F_9$, $F_{11}$) out of the thirteen test functions. In ($F_1$, $F_2$, $F_4$, $F_6$, $F_7$, $F_{12}$, $F_{13}$) test functions, the RESHWOA did not find the optimal solution, but they are very close to theoretical optimal solutions, except in four ($F_3$, $F_5$, $F_8$, $F_{10}$). In the second part, as the dimension increases, the RESHWOA also has precedence in ($F_1$, $F_2$, $F_4$, $F_6$, $F_7$, $F_9$, $F_{12}$, $F_{13}$), particularly in the function $F_9$, See Table 4. We calculated the standard deviation for each set of runs and recorded the results. Moreover, the minimal level of the Std. indicates how the algorithm showed the best result near the global value. RESHWOA showed consistent behaviour for all the test functions that the least values of the Std. can judge. Tables 3 and 4 also provide the average time of the algorithm.

**3.1.5 Acceleration convergence experiment.**    We thoroughly examine the convergence behaviour of the algorithm by plotting its optimal values in each iteration for dimension thirty. Figs 3–15 depict the convergence graphs of the two techniques for each of the thirteen benchmark test functions. RESHWOA achieves the most rapid and significant convergence for four distinct types of tests, namely $F_3$, $F_6$, $F_8$ and $F_{12}$. The suggested algorithm took first place and performed exceptionally well in terms of quicker convergence to the best value of the test

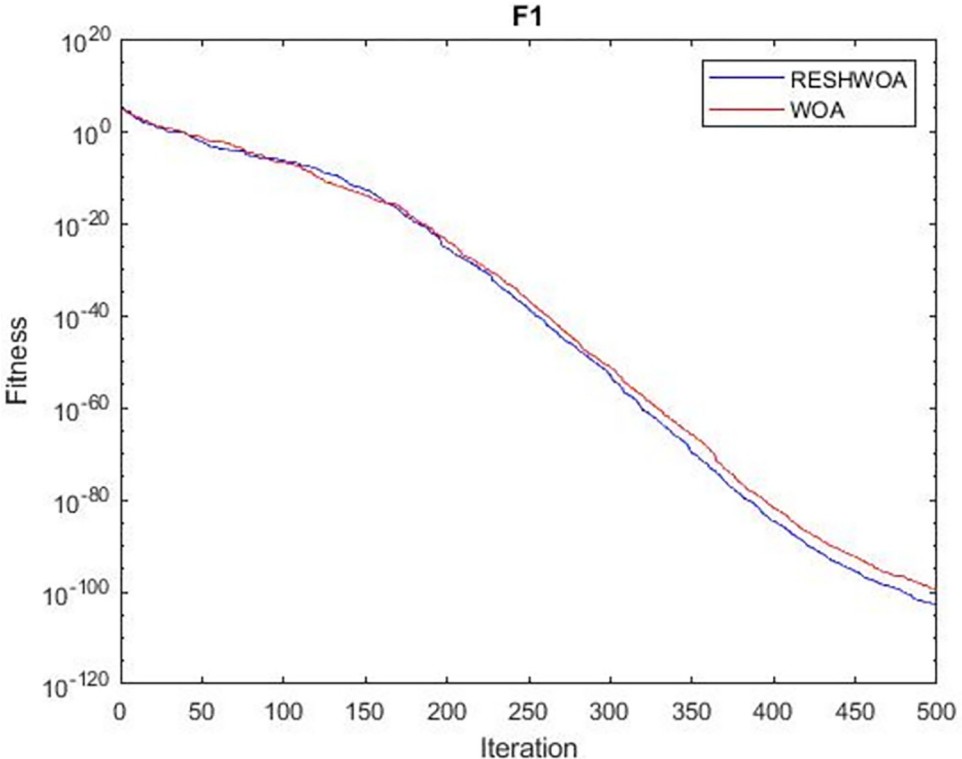

**Fig 3. WOA and RESHWOA convergence curves.**

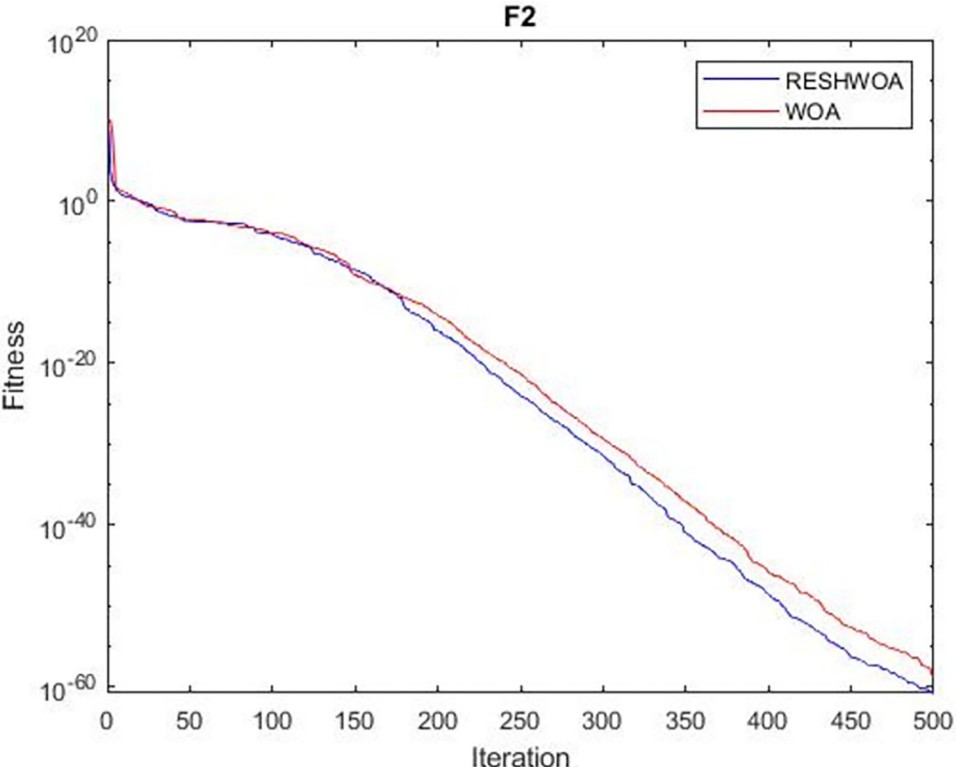

**Fig 4. WOA and RESHWOA convergence curves.**

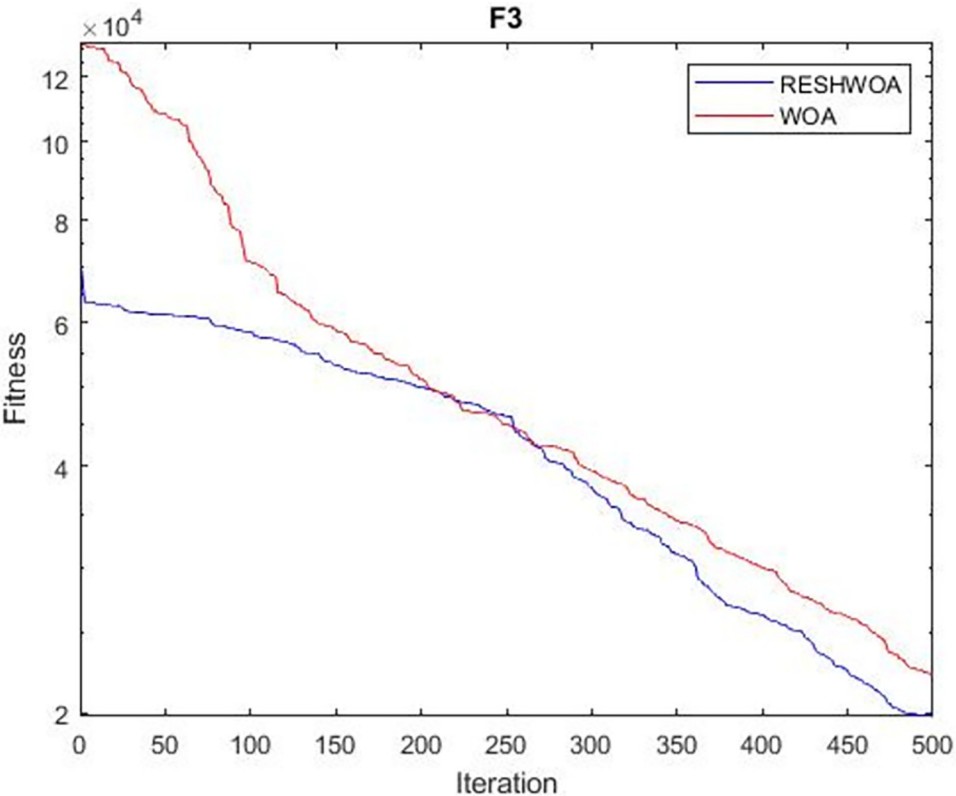

**Fig 5. WOA and RESHWOA convergence curves.**

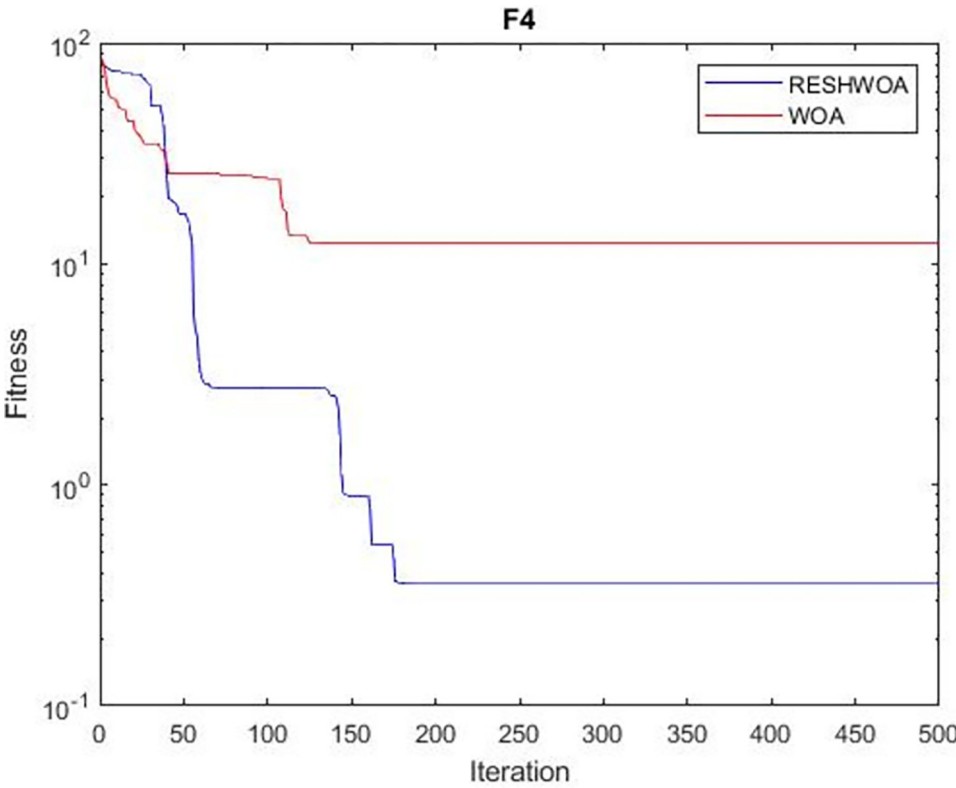

**Fig 6. WOA and RESHWOA convergence curves.**

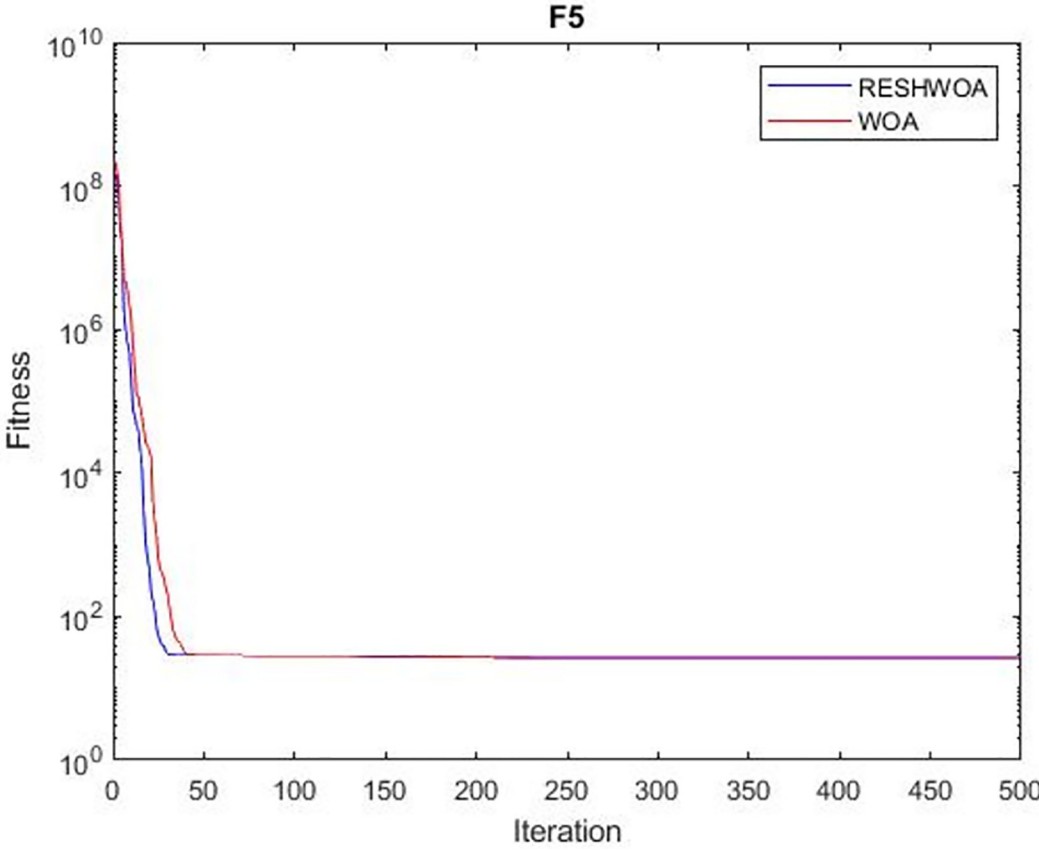

**Fig 7. WOA and RESHWOA convergence curves.**

function. Except for $F_{10}$, all test functions showed poor WOA convergence. When observing convergence behavior $F_1$, $F_2$, $F_5$, $F_9$ and $F_{11}$, we find that both algorithms exhibit equal convergence. We illustrated the results in Figs 3–15.

## 4. Algorithm performance on cancer data

A microarray is a laboratory instrument that detects the expression of thousands of genes at the same time. The main disadvantage of microarray data is the curse of dimensionality, which obstructs helpful information in a data set and causes computational instability. Therefore, relevant gene selection in microarray data analysis is complex [21]. In this study, we examined microarray data and the results of optimization.

### 4.1 Data sets

We used two datasets, one with 2,000 features (the smallest) and the other with 24,481 features (the largest). Table 5 details the datasets. We conducted the experiments using MATLAB 2020 on a Windows 10 platform running on an Intel Core i7 computer. We used two different approaches for feature selection to improve the model's performance.

**4.1.1 Filtration techniques and reduced data sets.** We apply two data reduction techniques before applying the proposed method. These techniques are good in avoiding redundant genes because they use information from normal and cancer genes.

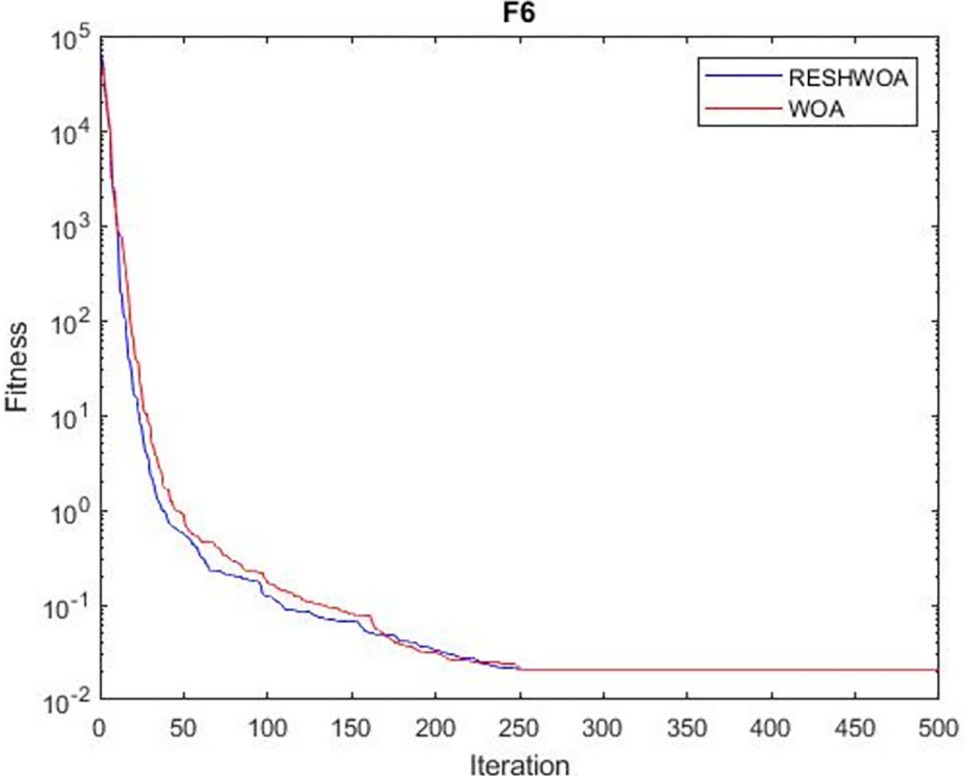

**Fig 8. WOA and RESHWOA convergence curves.**

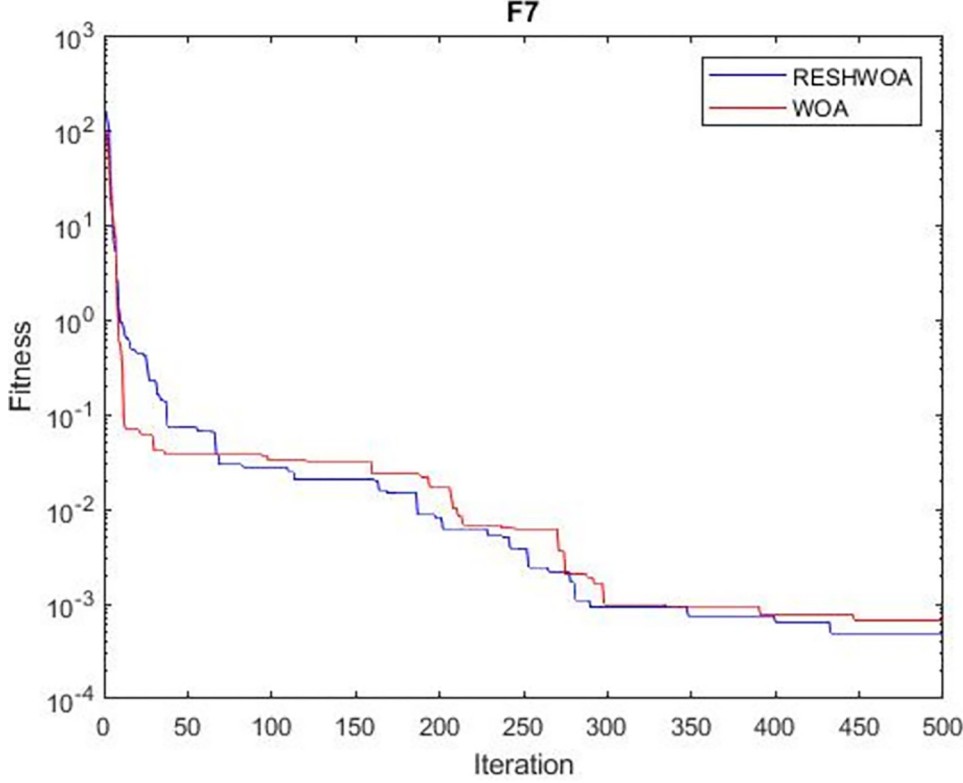

**Fig 9. WOA and RESHWOA convergence curves.**

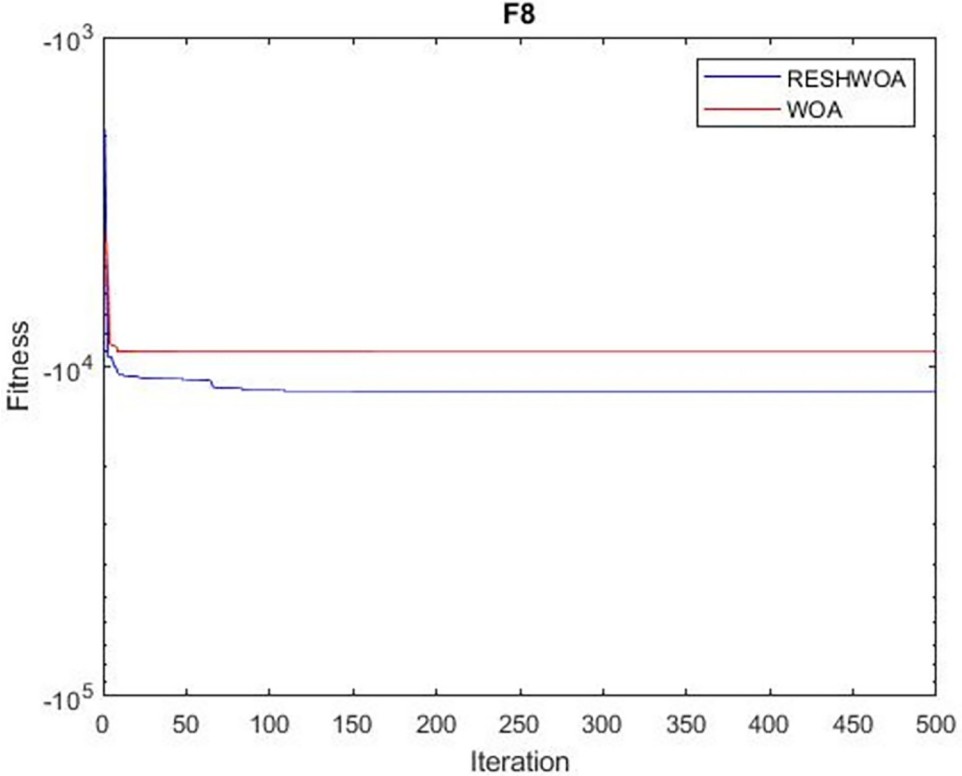

**Fig 10. WOA and RESHWOA convergence curves.**

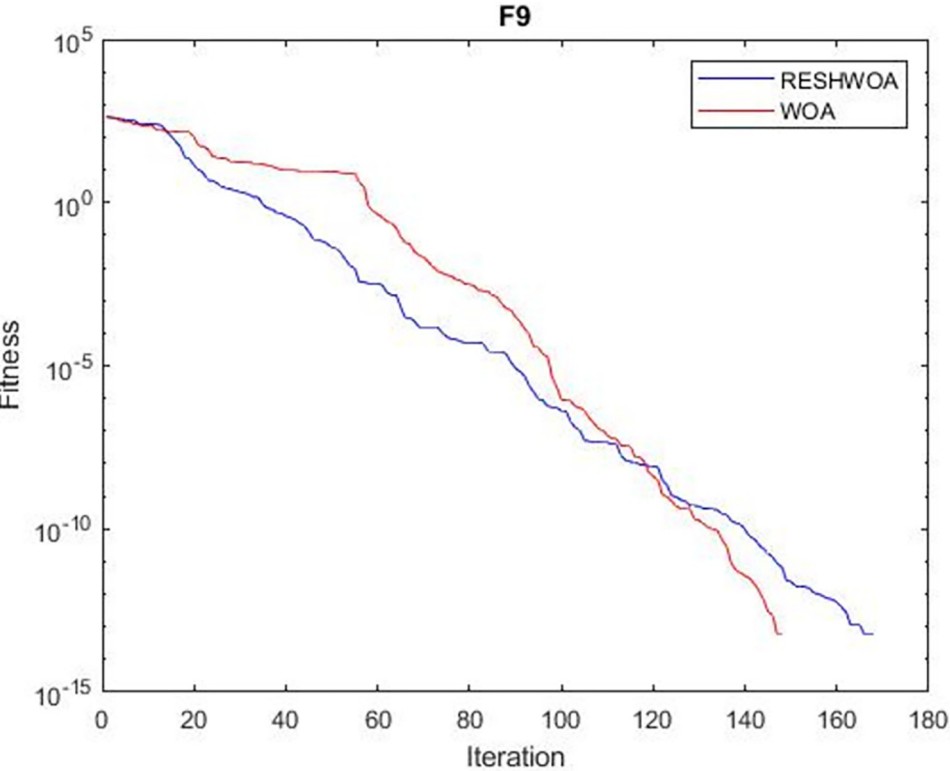

**Fig 11. WOA and RESHWOA convergence curves.**

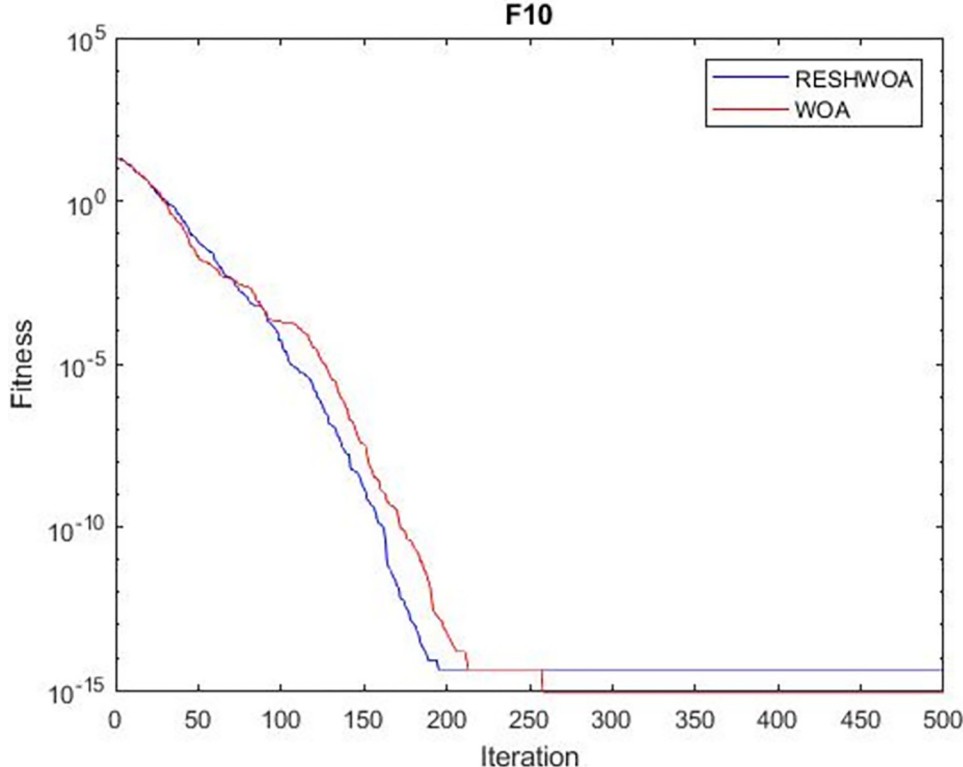

**Fig 12. WOA and RESHWOA convergence curves.**

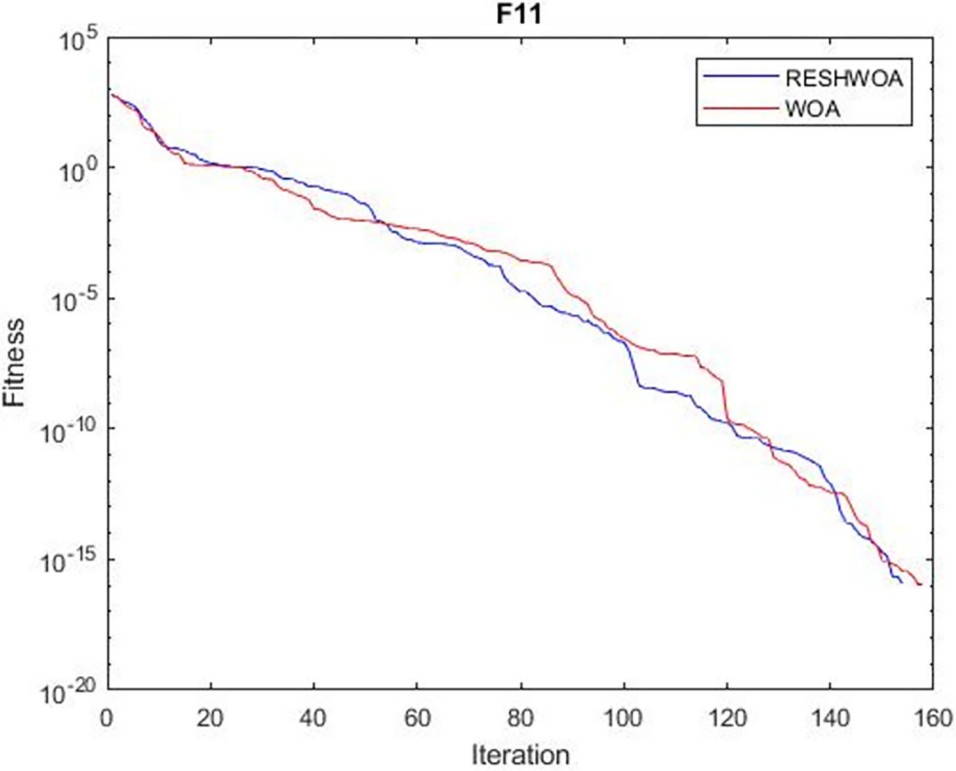

**Fig 13. WOA and RESHWOA convergence curves.**

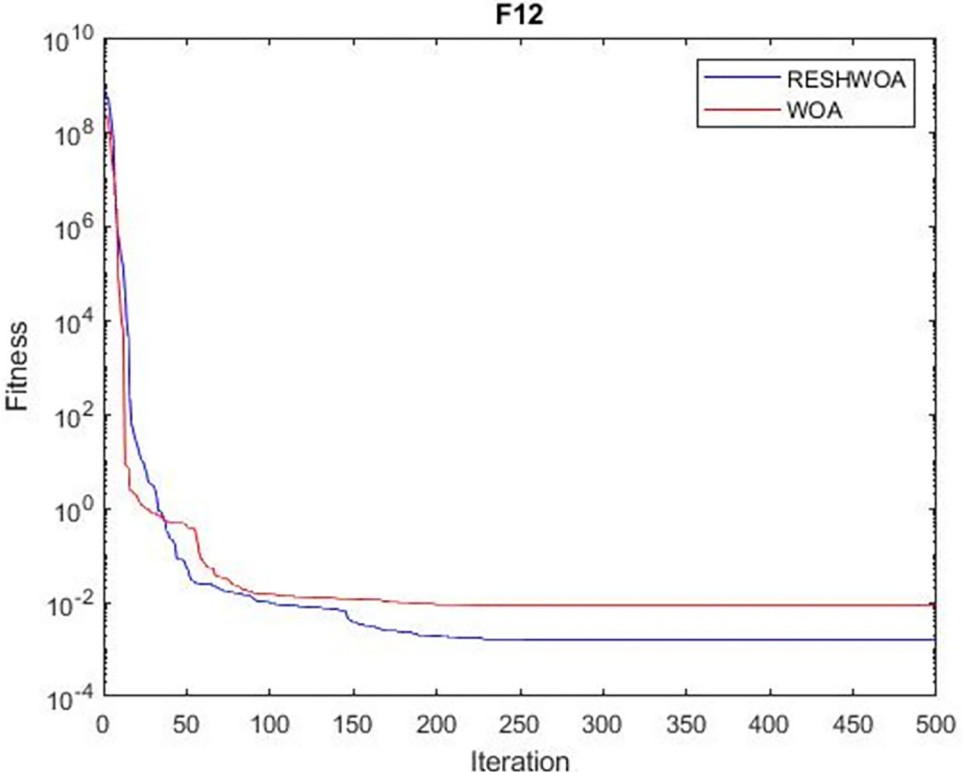

**Fig 14. WOA and RESHWOA convergence curves.**

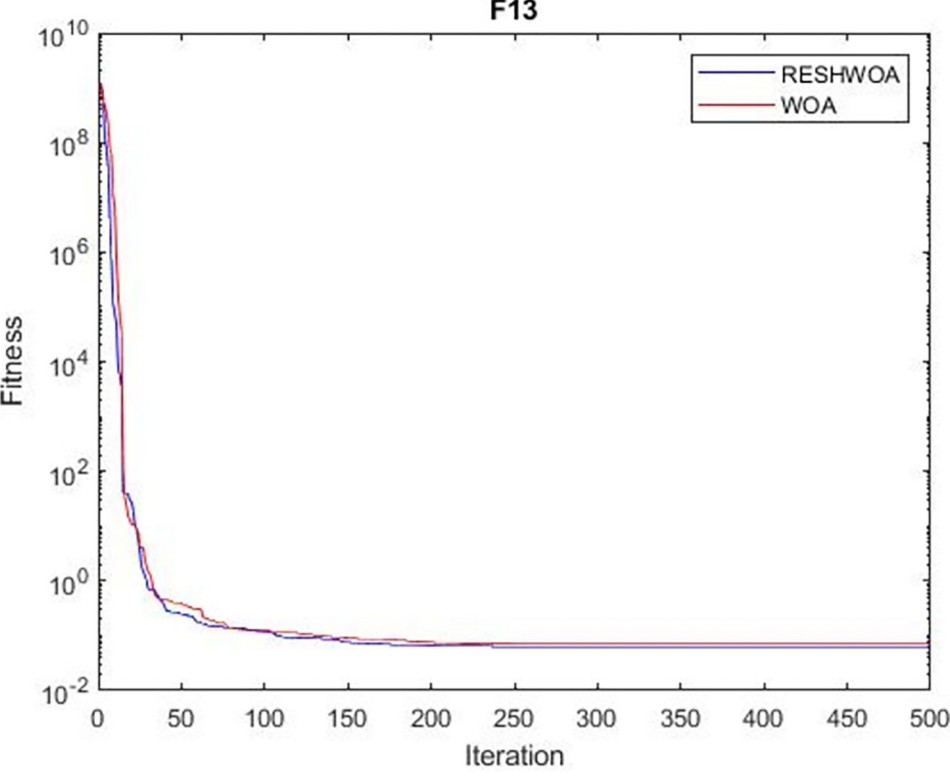

**Fig 15. WOA and RESHWOA convergence curves.**

**Table 1. Unimodal benchmark functions.**

| Function Name | Function | D | S | $F_{min}$ |
|---|---|---|---|---|
| Sphere | $F_1(X) = \sum_{i=1}^{D} X_i^2$ | 30,100 | $[-100, 100]^D$ | 0 |
| Schwefel's 2.22 | $F_2(X) = \sum_{i=1}^{D} \lvert X_i \rvert + \prod_{i=1}^{D} \lvert X_i \rvert$ | 30,100 | $[-10, 10]^D$ | 0 |
| Schwefel's 1.20 | $F_3(X) = \sum_{i=1}^{D} (\sum_{i=1}^{D} X_i)^2$ | 30,100 | $[-100, 100]^D$ | 0 |
| Schwefel's 2.21 | $F_4(X) = \max\{\lvert X_i \rvert, 1 \leq i \leq D\}_i$ | 30,100 | $[-100, 100]^D$ | 0 |
| Rosenbrock | $F_5(X) = \sum_{i=1}^{D} [100(X_{i+1} - X_i^2)^2 + (X_i - 1)^2]$ | 30,100 | $[-30, 30]^D$ | 0 |
| Step | $F_6(X) = \sum_{i=1}^{D} (X_i + 0.5)^2$ | 30,100 | $[-100, 100]^D$ | 0 |
| Quartic Noise | $F_7(X) = \sum_{i=1}^{D} iX_i^4 + random[0, 1)$ | 30,100 | $[-1.28, 1.28]^D$ | 0 |

**Table 2. Multimodal benchmark functions.**

| Function Name | Function | D | S | $F_{min}$ |
|---|---|---|---|---|
| Schewefel's 2.26 | $F_8(X) = \sum_{i=1}^{D} -X_i \sin(\sqrt{\lvert X_i \rvert})$ | 30,100 | $[-500, 500]^D$ | -148.9829*n |
| Rastrigin | $F_9(X) = [X_i^2 - 10\cos(2\pi X_i + 10)]$ | 30,100 | $[-5.12, 5.12]^D$ | 0 |
| Ackley | $F_{10}(X) = -20exp\left(-0.2\sqrt{\frac{1}{D}\sum_{i=1}^{D} X_i^2}\right)$ $-exp\left(\frac{1}{D}\sum_{i=1}^{D}\cos(2\pi X_i)\right) + 20 + e$ | 30,100 | $[-32, 32]^D$ | 0 |
| Griewank | $F(X) = \frac{1}{400}\sum_{i=1}^{D} X_i^2 - \prod_{i=1}^{D}\cos\left(\frac{X_i}{\sqrt{i}}\right) + 1$ | 30,100 | $[-600, 600]^D$ | 0 |
| Pendlized | $F_{12}(X) = \sum_{i=1}^{D} U(X_i, 10, 100, 4)$ $+\frac{\pi}{D}\{10sin^2(3\pi y_i) + \sum_{i=1}^{D-1}(y_i - 1)^2[1 + sin^2(3\pi y_{i+1})] + (y_D - 1)^2\}$ $y_i = 1 + \frac{1}{4}(X_i + 1)$ $U(X_i, a, k, m) = \begin{cases} k(X_i - 1)^m, & X_i > a, \\ 0, & -a \leq X_i \leq a, \\ k(-X_i - 1)^m, & X_i < -a, \end{cases}$ | 30,100 | $[-50, 50]^D$ | 0 |
| Generalized Pendlized | $F_{13}(X) = \sum_{i=1}^{D} U(X_i, 10, 100, 4)$ $+\frac{1}{10}\{sin^2(3\pi X_i) + \sum_{i=1}^{D-1}(X_i - 1)^2[1 + sin^2(3\pi X_{i+1})] + (X_D - 1)^2[1 + sin^2(2\pi X_D)]\}$ $U(X_i, a, k, m) = \begin{cases} k(X_i - 1)^m, & X_i > a, \\ 0, & -a \leq X_i \leq a, \\ k(-X_i - 1)^m, & X_i < -a, \end{cases}$ | 30,100 | $[-50, 50]^D$ | 0 |

**Table 3. Test results of the benchmark functions, the dimension fixed to thirty.**

| Function | WOA | | | RESHWOA | | | |
|---|---|---|---|---|---|---|---|
| | Avg. | Std. | Time | Avg. | Std. | Avg. Time | benchmark |
| F1 | 1.0683e-74 | 3.0191e-74 | 01.2780 | **1.8256e-10** | 6.9650e-10 | 05.1139 | **0** |
| F2 | 1.4999e-49 | 6.8637e-49 | 01.4028 | **5.9813e-60** | 2.2459e-59 | 05.1934 | **0** |
| F3 | 4.2659e+04 | 1.3960e+04 | 05.9696 | 1.5931e+04 | 7.9631e+03 | 21.5713 | 0 |
| F4 | 47.4474 | 25.6257 | 01.5514 | **18.3155** | 26.0905 | 05.0813 | 0 |
| F5 | 27.9464 | 0.4439 | 01.8927 | 27.0641 | 0.6092 | 06.8722 | 0 |
| F6 | 0.3906 | 0.2617 | 01.2809 | **0.0437** | 0.0996 | 05.0578 | 0 |
| F7 | 0.0015 | 0.0015 | 03.8390 | **9.8461e-04** | 0.0014 | 13.5069 | 0 |
| F8 | -1.0523e+04 | 1.7622e+03 | 01.8925 | -1.1591e+04 | 1.2476e+03 | 07.0700 | -418.9829*30 |
| F9 | 3.7896e-15 | 2.0756e-14 | 01.5322 | **0** | 0 | 05.3827 | **0** |
| F10 | 3.6119e-15 | 2.9033e-15 | 01.5966 | 4.0856e-15 | 1.9459e-15 | 05.7373 | 0 |
| F11 | 0.0043 | 0.0236 | 01.9610 | **0** | 0 | 07.1504 | **0** |
| F12 | 0.0286 | 0.0240 | 08.5613 | **0.0048** | 0.0086 | 29.7863 | 0 |
| F13 | 0.4524 | 0.2052 | 08.6047 | **0.1038** | 0.1104 | 29.3452 | 0 |

*4.1.1.1 Signal-to-noise ratio (SNR).* The signal-to-noise ratio test identifies the gene expression pattern with a significant difference in mean and variance within each group [62]. We select the top-ranked genes through the SNR test statistics according to their expression levels. Below, we provide the formula for the method.

$$\text{Signal}-\text{to}-\text{noise ratio} = \frac{\mu_1 - \mu_2}{\sigma_1 + \sigma_2} \tag{10}$$

Where $\mu_1$ and $\mu_2$ are the mean expression values for the sample, classes 1 and 2, respectively. The standard deviations are $\sigma_1$ and $\sigma_2$ in each class.

To discover differentially expressed genes with SNR, we perform the following steps.

**Step 1**: Normalize the data.

**Step 2**: Separate the two groups for normal and disease data.

**Step 3**: Evaluate the signal-to-noise ratio.

**Table 4. Test results of the benchmark function, the dimension fixed to one hundred.**

| Function | WOA | | | RESHWOA | | | |
|---|---|---|---|---|---|---|---|
| | Avg. | Std. | Avg. Time | Avg. | Std. | Avg. Time | benchmark |
| F1 | 1.8822e-69 | 1.0299e-68 | 02.7470 | **4.1499e-100** | 1.3019e-99 | 03.7140 | **0** |
| F2 | 3.7181e-48 | 1.6884e-47 | 02.7874 | **7.7554e-58** | 2.4461e-57 | 03.2223 | **0** |
| F3 | 1.0457e+06 | 3.8371e+05 | 26.0376 | 6.5064e+05 | 1.4678e+05 | 91.0296 | 0 |
| F4 | 67.2380 | 28.5696 | 02.7059 | **62.7189** | 35.6743 | 08.9364 | 0 |
| F5 | 98.2406 | 0.2069 | 03.3927 | 97.4164 | 0.4801 | 12.3029 | 0 |
| F6 | 4.1325 | 1.0834 | 02.5921 | **1.1605** | 0.5098 | 09.1404 | 0 |
| F7 | 0.0034 | 0.0039 | 11.1243 | **0.0012** | 0.0018 | 38.1394 | 0 |
| F8 | -3.6407e+04 | 6.1080e+03 | 04.4031 | -3.9287e+04 | 3.1190e+03 | 14.9302 | -418.9829*100 |
| F9 | 0 | 0 | 03.0551 | **0** | 0 | 10.3598 | **0** |
| F10 | 4.7962e-15 | 2.8529e-15 | 03.2018 | 4.3225e-15 | 2.5523e-15 | 11.0596 | 0 |
| F11 | 3.7007e-18 | 2.0270e-17 | 03.990081 | 0.0069 | 0.0264 | 13.8847 | **0** |
| F12 | 0.0576 | 0.0372 | 23.853848 | **0.0102** | 0.0056 | 79.2452 | 0 |
| F13 | 2.7122 | 0.6399 | 24.335468 | **0.9871** | 0.5980 | 80.5072 | 0 |

**Table 5. Information of data sets and reduced data by data reduction techniques BC and SNR.**

| S. No. | Data sets | Classes | Sample | Total Genes | BC-Reduced data sets | SNR-Reduced Data sets | Ratio of BC | Ratio of SNR |
|---|---|---|---|---|---|---|---|---|
| 1 | Breast | Relapse(46) non_relapse (51) | 97 | 24481 | 23 | 20 | 0.09% | 0.08% |
| 2 | Carcinoma | Tumour(18),Normal(18) | 36 | 7464 | 62 | 20 | 0.83% | 0.27% |
| 3 | Colon | Tumour(40),Normal(22) | 62 | 2000 | 06 | 20 | 0.30% | 1.0% |
| 4 | CNS | Tumour(21),Normal(39) | 60 | 7129 | 36 | 20 | 0.50% | 0.28% |
| 5 | Ovarian | Normal(91),Cancer(162) | 253 | 15154 | 24 | 20 | 0.16% | 0.13% |
| 6 | Leukemia | ALL (47), AML (25) | 72 | 7129 | 28 | 20 | 0.39% | 0.28% |
|  | Average FF |  |  |  |  |  | 0.38% | 0.34% |

**Step 4**: Sort the data in ascending order.

**Step 5:** Select the top twenty genes.

**Step 6:** Fed this selected gene to the SVM.

**Step 7**: We use MSE as the objective function of the proposed method.

*4.1.1.2 Bhattacharyya distance (BC).* Bhattacharyya distance is a metric to evaluate the similarity between two probability distributions [71]. No previous investigation exists on feature selection using BC distance, and our proposed method has found utility in the optimum choice of features [72].

$$B = \frac{1}{8}(\mu_i - \mu_j)^t \left(\frac{\sigma_i + \sigma_j}{2}\right)^t (\mu_i - \mu_j) + \frac{1}{2} ln \frac{|\frac{\sigma_i + \sigma_j}{2}|}{\sqrt{(|\sigma_i||\sigma_j|)}} \qquad (11)$$

$\mu_i$ and $\sigma_i$ refer to the mean and variance of the gene in the cancer sample, $\mu_j$ and $\sigma_j$ refer to the mean and variance of the gene in the normal tissue samples. The greater the distance is, the stronger the relationship between genes and cancer. Following are the steps for the Bhattacharyya distance.

**Step 1:** We have a set of genes S = {F1, F2, F3. . .. . .. . .Fn}.

**Step 2:** Evaluate the Bhattacharyya distance (BC).

**Step 3:** Sort the distance in descending order. (i.e., the values with maximum dissimilarity)

**Step 4:** We determine the benchmark value through a hit-and-trial method.

**Step 5:** Select the genes based on the threshold value if BC>threshold.

**Step 6:** Subset of the informative genes are {F1, F2, F3,. . .Fs}

We investigated marker genes using BC and SNR techniques and presented the selected genes in Table 5. We employed the hit-and-trial method to determine the threshold value of BC. On the other hand, we set the top twenty genes using SNR. Since these methods only select a limited subset of informative genes, we used reduced datasets in all our experiments below to ensure we could conduct a rigorous analysis with a manageable number of features.

**4.1.2 Classifier and validation.** *4.1.2.1 Support vector machine.* The support vector machine is well-known for its high generalization capability and robustness when dealing with high-dimensional data [1]. Our SVM models use the radial basis function (RBF) kernel, which requires two parameters, penalty term C and kernel parameter $\gamma$, since these parameters directly impact classification performance [31]. Optimizing parameters is crucial for achieving better classification accuracy. However, this process can be challenging and costly, and it may compromise the reliability of the results. While the default parameters of support vector machines (SVM) can yield satisfactory performance, fine-tuning them through parameter optimization can significantly improve classification accuracy [73]. The lower and upper limits of *c* are 0.0001 and 100, while for $\gamma$, lower and upper limits are 0.001 and 50, respectively. The trial-and-error approach defines these bounds. After initialization, we evaluate the search

space for each candidate with an objective function. This study uses the mean squared error (MSE) as the objective function.

The objective function's primary purpose is to determine the individual within the population with the lowest loss function value, as this member is regarded as the best performer among all individuals. We must minimize the objective function through the optimization algorithm to obtain the optimal solution and optimize the parameters $c$ and $\gamma$. Subsequently, we assessed the accuracy of the proposed method and WOA regarding mean squared error (MSE) as the evaluation metric. Cross-validation is a technique to reduce the problem of overfitting. To choose parameters $c$ and $\gamma$ using the holdout method, we split data into seventy by thirty ratios for training and testing purposes. We use one subset to train the model and another to evaluate the predictions.

We apply the SVM method with the following presets.

1. We use the Radial basis function (RBF) kernel.

2. We split the data into training and testing using holdout.

3. We optimize the kernel scale and box constraints.

*4.1.2.2 Validation.* We use the data set $S = \{(X_i, Y_i)|i \in N\}$ of microarray cancer data with the defined binary class diseased or non-diseased to assess predictive accuracy. We divided the data into two disjoint subsets for training and testing purposes, i.e., $S_{train} \cup S_{test} = S$ and $S_{train} \cap S_{test} = \emptyset$. We integrated an SVM model with measured responses on the training subset $S_{train}$ and used it to estimate the unknown responses in the test subset $S_{test}$.

Adding the predictions from the disjoint test sets, as a result, for the original data set S, we now have out-of-sample predictions. Two aspects are critical for model validation: discrimination and calibration. Discrimination affects only classification, whereas calibration affects both classification and accuracy. Discrimination assesses the model's ability to distinguish between high and low-risk individuals without considering the absolute values of the predictions. Conversely, calibration quantifies the similarity of predicted outcome variables or ratings to observed outcomes. We want to predict the SVM parameter's accuracy as precisely as possible in our classification task. As a result, we require well-calibrated models. The mean squared error (MSE), which we normalized by dividing by the sample size, is a statistic that measures both discrimination and calibration. We confirmed that the model produced consistent results across all datasets. Mean Squared Error (MSE) quantifies and communicates the model's accuracy.

$$MSE(S_{test}) = \frac{1}{|S_{test}|} \sum_{i \epsilon S_{test}} (y_i - \hat{y}_i)^2$$

*4.1.2.3 Feature encoding for SVM training.* Our study used a machine learning approach to analyze the data. Specifically, we employed a Support Vector Machine (SVM) classifier to perform classification tasks. To prepare the data for SVM training, we implemented a feature encoding process. This process involved converting categorical labels into numerical values, which are suitable for SVM training. We performed the feature encoding as follows. To illustrate our algorithm's operation, please refer to Fig 16, which provides a visual representation of how these encoded features are utilized.

1. **Loading the Data:** We load the data from the 'file_name.xlsx' Excel file using the 'readtable' function.

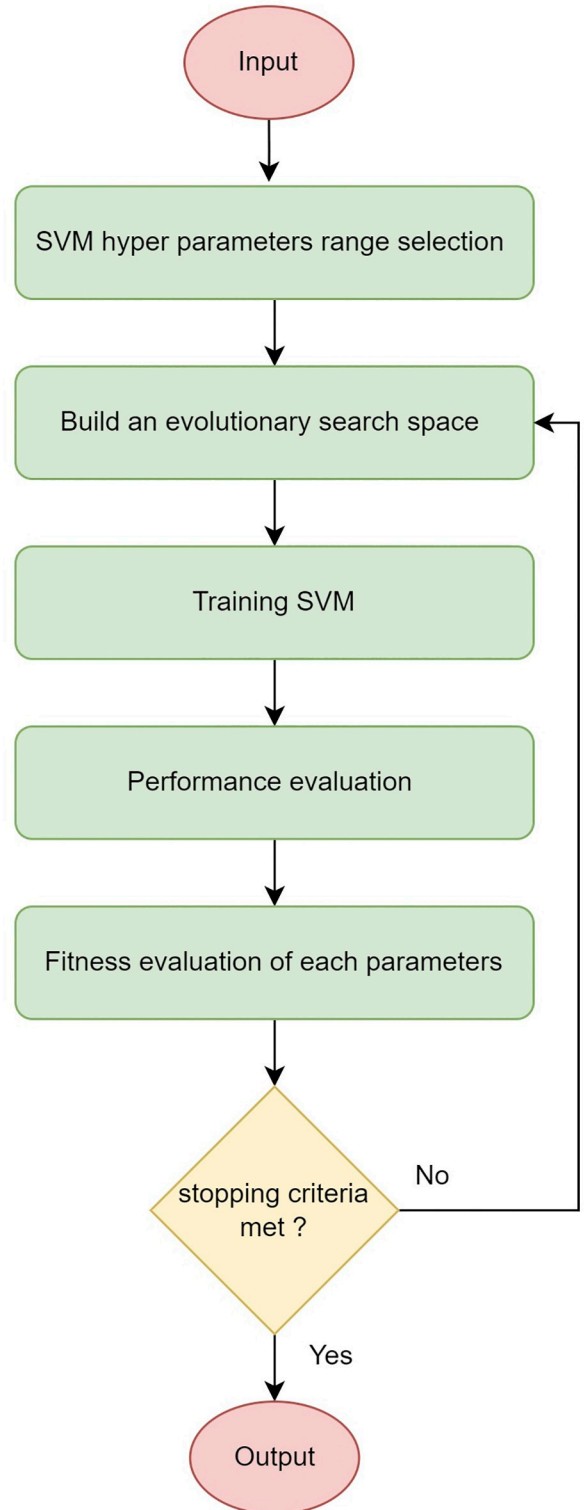

**Fig 16. Flow diagram of the proposed algorithm.**

2. **Defining Categorical Labels and Numerical Values:** We define the categorical labels in the 'k' array as ["Tumour", "Normal"], and we define their corresponding numerical values in the 'l' array as [1, 0]. These arrays establish a mapping between the categorical labels and their numerical representations.

3. **Encoding Features**: The loop iterates through the 'y' column of the data (assumed to contain the categorical labels). For each label in the 'y' column, it checks if it matches either "Tumour" or "Normal" (as defined in 'k'). If we find a match, it assigns the corresponding numerical value (1 or 0) to the 'number' array. This process encodes the categorical labels as numerical values.

4. **Adding Encoded Features to the Data**: We add the encoded labels to the data as a new column named 'category_encoded'. This column contains the numerical representations of the categorical labels.

Following these steps, we transform the original categorical features in the 'y' column of the data into a numerical format suitable for training the SVM model. This encoding allows the SVM model to work with the data effectively.

## 4.2 SVM Parameter optimization using RESHWOA and WOA

To optimize the SVM parameters $c$ and $\gamma$, we used the traditional WOA and proposed RESHWOA. with ub = [10 100], lb = [0.0001 0.1], and runs = 30.

**4.2.1 SNR results of RESHWOA and WOA.** The proposed model achieved lower averages in all datasets than the WOA algorithm, as shown in Table 6. Furthermore, the proposed model provided minimum standard deviation measures in all datasets except two; however, WOA performance in two datasets, carcinoma, and colon, is good. In all datasets, RESHWOA outperforms as compared to WOA in terms of minimum score except in the Colon and Breast cancer datasets. Overall, the proposed method significantly improved performance in the presence of the SNR. This experiment demonstrates that the RESHWOA always finds a near-optimal parameter combination with minimum MSE in the given range. We highlight the best values in bold.

**4.2.2 BC results of RESHWOA and WOA.** In the previous experiment with Colon and Breast cancer datasets, the signal-to-noise ratio failed to identify the minimum MSE score. In contrast, BC gives the minimum error score 'zero' in all data sets in the current investigation except for CNS data because the minimum error score is 0.5%. Furthermore, the proposed model provided minimum standard deviation measures in all datasets except Carcinoma and Leukemia, while WOA performs best in both. The proposed method yielded the best parameter combination with the lowest MSE score in most data. Table 7 displays the efficiency of the proposed algorithm.

**Table 6. SNR results of RESHWOA and WOA with thirty runs and fifty iterations.**

| S. No | Data sets | RESHWOA parameters (C, $\gamma$) at min MSE | RESHWOA Min MSE | RESHWOA Avg. MSE | RESHWOA Avg. Std | WOA SVM Parameters (C, $\gamma$) at min MSE | WOA Min MSE | WOA Avg. MSE | WOA Avg. Std |
|---|---|---|---|---|---|---|---|---|---|
| 1 | Carcinoma | 0.923, 73.933 | **0.0000** | **0.0013** | 0.0024 | 5.031,20.832 | 0.5200 | 0.5200 | **3.3876e-16** |
| 2 | CNS | 1.878, 75.967 | **0.0000** | **3.7665e-04** | **0.0014** | 5.328, 25.214 | 0.2857 | 0.2865 | 0.0043 |
| 3 | colon | 2.672, 89.281 | **0.0056** | **0.0019** | 0.0027 | 7.275, 52.180 | 0.3023 | 0.3023 | **5.6460e-17** |
| 4 | Breast | 4.970, 95.967 | <u>**0.1194**</u> | **0.1095** | **0.0120** | 10, 100 | <u>**0.1194**</u> | 0.1527 | 0.0202 |
| 5 | Leukemia | 6.616, 88.078 | **0.0000** | **5.6497e-04** | **0.0017** | 0.000, 0.1028 | 0.2000 | 0.2767 | 0.0167 |
| 6 | Ovarian | 7.543, 23.674 | <u>**0.0000**</u> | **0.0011** | **0.0023** | 7.549, 34.188 | <u>**0.0000**</u> | 0.0119 | 0.0045 |

**Table 7. BC results of RESHWOA and WOA with thirty runs and fifty iterations.**

| S. No. | Data Sets | RESHWOA SVM parameters (C, $\gamma$) at min MSE | RESHWOA Min MSE | RESHWOA Avg. MSE | RESHWOA Avg. Std | WOA SVM parameters (C, $\gamma$) at min MSE | WOA Min MSE | WOA Avg. MSE | WOA Avg. Std |
|---|---|---|---|---|---|---|---|---|---|
| 1 | Carcinoma | 6.7891, 100 | **0** | **3.7665e-04** | 0.0014 | 9.267, 11.566 | **0** | 0 | 0 |
| 2 | CNS | 10, 100 | **0.0056497** | **7.5330e-04** | **0.0020** | 4.422, 24.812 | 0.11905 | 0.1587 | 0.0201 |
| 3 | colon | 10, 100 | **0** | **7.5330e-04** | **0.0020** | 9.498,92.920 | 0.046512 | 0.1023 | 0.0233 |
| 4 | Breast | 10, 34.157 | **0** | **3.7665e-04** | **0.0014** | 0.169, 63.283 | 0.37313 | 0.4159 | 0.0217 |
| 5 | Leukemia | 2.7157, 56.666 | **0** | **3.7665e-04** | 0.0014 | 6.748, 49.153 | **0** | 0 | 0 |
| 6 | Ovarian | 2.2702, 40.578 | **0** | **9.4162e-04** | **0.0021** | 4.296, 71.344 | **0** | 0.0083 | 0.0038 |

Table 8 summarizes the previous work done for parameter optimization where optimization techniques and kernel functions of SVM vary [32, 74–76]. All these kernels are acceptable because there are specific issues for which each of them is the best option. We have listed fourteen publicly available datasets from published studies in Table 8 and the number of optimization techniques used for each dataset. To our knowledge, no existing literature currently utilizes WOA as a parameter optimization technique. Using an improved version of the WOA, we optimized the SVM parameter by BC and SNR integrated techniques. From the results, RESHWOA features have improved minimum error over the existing method; this is a significant achievement. Fig 17 indicates the performance comparison of BC and SNR. Both techniques showed a good result, but BC indicates a more optimal result than SNR.

## 5. Conclusion

This paper proposes a hybridization algorithm of the whale optimization algorithm (WOA) with a discrete recombinative (DR) strategy. The improved RESHWOA algorithm adds a diverse random population and tends to the global optimum, which enhances the global search capability of the RESHWOA algorithm. The RESHWOA algorithm draws inspiration from the phase exchanging information found in the DR strategy, and it introduces two control parameters, μ and ρ. These parameters play a crucial role in guiding the algorithm's behaviour. To evaluate the performance of the RESHWOA algorithm, we assessed it on thirteen benchmark test functions. We then compared the results of this evaluation with those obtained from the state-of-the-art WOA algorithm. The aim of this comparison is to assess the performance of the RESHWOA algorithm in relation to the existing advanced optimization technique WOA. The results show that the RESHWOA algorithm has better global exploration

**Table 8. Typical datasets and parameter optimization techniques with a specific kernel.**

| Datasets (High Dimensional) | Optimization techniques | Kernel (SVM) | Impression using the optimization algorithm | Authors |
|---|---|---|---|---|
| Leukemia, Embryonal Tumours, Dexter, Internet ads, Madelon, Musk, Spam base, SPECTF Heart, Intrusion | Grid search, GA. | Linear, Rbf, Polynomial, Sigmoid | GA has proven to be more stable than Grid search. | Iwan Syarif (2016) |
| Acute leukemia, Breast cancer | GA. | Gaussian | Perform well in gene selection and achieve high classification accuracies with a small number of genes. | Mao Yong (2005) |
| GT data | ACO, GA ICA, PSO | RBF | We improved all five accuracy criteria from a confusion matrix by at least 5% compared to SVM. | Elahe Tamimi (2017) |
| PIMA, WDBC | PSO, DE, HS, ABC, TLBO | RBF | TLBO outperforms various SVM model selection methods proposed in the literature, particularly the well-known Bayesian method. | Ghnimi (2020) |

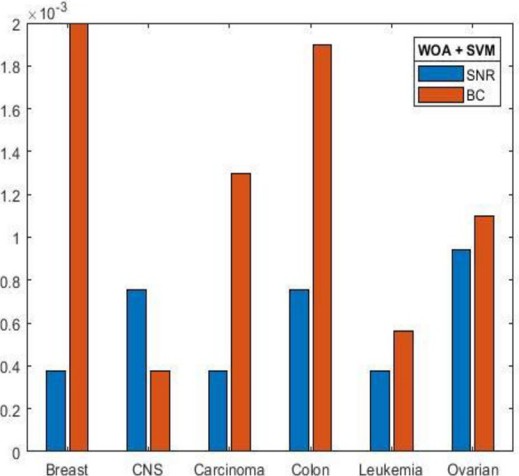 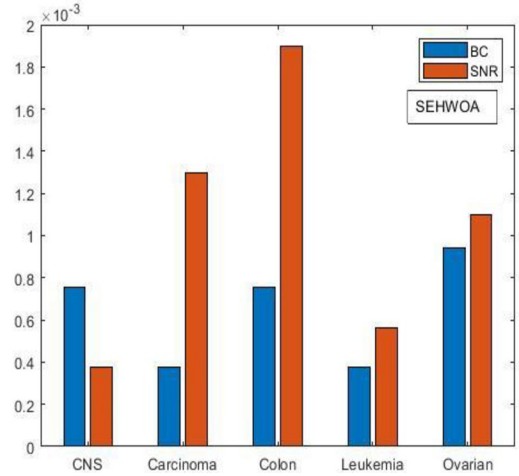

**Fig 17. Accuracy of SVM with SNR and BC for different data sets.**

capabilities and higher convergence accuracy. Furthermore, the SVM parameter optimization experimental results, using SNR and BC on microarray cancer data, show the best performance in five data sets out of six. This experiment demonstrates that the RESHWOA offers a near-optimal parameter combination with minimum MSE in the given range. Although the proposed RESHWOA algorithm is a hybridization, the detailed simulation experiments in this paper verified its better performance. The iterative mechanism effectively enhances the randomness in the control parameter. Various fields can apply this approach and may have significant implications for future research. Building upon these promising findings, several potential research directions emerge for future investigation: i) integration with other metaheuristic algorithms, ii) Further exploration of DR strategy, iii) parameter tuning and sensitivity analysis of DR strategy, iv) How to readily hybridize existing algorithms with fast convergence, low MSE, stability, and steadiness might be promising work in the future.

## Supporting information

**S1 Fig. Description of evolutionary strategy.**
(TIF)

## Acknowledgments

The authors are grateful to the editor and anonymous reviewers for their constructive comments and valuable suggestions, which not only improved the quality of the paper but added value to it.

## Author Contributions

**Supervision:** Sana Saeed.

**Writing – original draft:** Rahila Hafiz.

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
