## [Decision Letter · Decision Letter 0]

9 Feb 2023

PONE-D-23-01358A state-of-the-art Fusion of Whale algorithm with Evolutionary Strategies for high dimensionality and Filtration with Signal to Noise Ratio and Bhattacharyya DistancePLOS ONE

Dear Dr. hafiz,

Thank you for submitting your manuscript to PLOS ONE. After careful consideration, we feel that it has merit but does not fully meet PLOS ONE’s publication criteria as it currently stands. Therefore, we invite you to submit a revised version of the manuscript that addresses the points raised during the review process.

We look forward to receiving your revised manuscript.

Kind regards,

Omar A. Alzubi

Academic Editor

PLOS ONE

Journal Requirements:

"No"

"No"

5. We note that you are reporting an analysis of a microarray, next-generation sequencing, or deep sequencing data set. PLOS requires that authors comply with field-specific standards for preparation, recording, and deposition of data in repositories appropriate to their field. Please upload these data to a stable, public repository (such as ArrayExpress, Gene Expression Omnibus (GEO), DNA Data Bank of Japan (DDBJ), NCBI GenBank, NCBI Sequence Read Archive, or EMBL Nucleotide Sequence Database (ENA)). In your revised cover letter, please provide the relevant accession numbers that may be used to access these data. For a full list of recommended repositories, see http://journals.plos.org/plosone/s/data-availability#loc-omics or http://journals.plos.org/plosone/s/data-availability#loc-sequencing.

6. Please ensure that you refer to Figures 1, 4 to 14 in your text as, if accepted, production will need this reference to link the reader to the figure.

7. We note you have included a table to which you do not refer in the text of your manuscript. Please ensure that you refer to Table 6 in your text; if accepted, production will need this reference to link the reader to the Table.

Reviewers' comments:

Reviewer's Responses to Questions

**Comments to the Author**

1. Is the manuscript technically sound, and do the data support the conclusions?

Reviewer #1: Partly

Reviewer #2: Partly

2. Has the statistical analysis been performed appropriately and rigorously? 

Reviewer #1: No

Reviewer #2: I Don't Know

3. Have the authors made all data underlying the findings in their manuscript fully available?

Reviewer #1: Yes

Reviewer #2: Yes

4. Is the manuscript presented in an intelligible fashion and written in standard English?

Reviewer #1: No

Reviewer #2: No

5. Review Comments to the Author

Reviewer #1: Large number of English language issues, including in abstract.

Needs extensive review by technical editor proficient in English.

for example the very first sentence is missing an 'IS'

'The standard whale algorithm IS easily trapped in suboptimal and high-dimensional regions.'

In many cases in past reviews I can still follow the flow of the document even when the English usage problems are significant; in this case, the paper is not well organized and it makes it harder to read such that both the organization and the English usage are difficult and problematic.

What is meant by trapped in subotimal regions? do you mean suboptimal solutions?

How is the algorithm trapped in high-dimensional regions? do you mean high dimensionality problems create problems finding optimal solutions?

Are there more references that could be used in section 2.2?

I would consider more references on WOA which give good context to nature-inspired and metaheuristics, such as

Gharehchopogh, Farhad Soleimanian, and Hojjat Gholizadeh. "A comprehensive survey: Whale Optimization Algorithm and its applications." Swarm and Evolutionary Computation 48 (2019): 1-24.

The WOA is not well explained; more background is needed.

The recombinative strategy description around line 167 is not well described at all and appears to mix generalities with specific parameters (5 parents, 100 offsping) that are of unclear origin.

An intuitive sense of the contrast between the collaborative nature of WOA and the recombination would be helpful. As I understand it, the WOA typically probabilistically updates each search agent; sometimes this uses the influence of a randomly selected search agent by using the influence of other search agents. The recombination is not clear as described in lines 154-174; the best I can understand is that the initial positions are found by recombinative methods rather than randomly, but it is not well described.

The presentation order is very hard to follow. The WOA is finally described in more detail starting at line 197, along with the updated version around line 220, but there is still no explanation of the number of parents / offspring which appear to be hard coded to some arbitrary number "100".

The solution appears to simply be selecting the initial population with the recombinative method, but it is not clear why this is better than simply having a better 'initial guess'.

what is the origin of the test functions in section 2.4.1? I do like the concept but are these functions that are used commonly in optimization problems particularly with WOA? If they were cited earlier than line 244, it would be helpful to repeat this ag line 244 "13 international test functions" (and what, exactly, is 'international' about a 'test function'?)

The table 1 says 'mean' and the document (line 247) says "Avg". The benchmark column is 0 for everything except F8 for dim=30 and dim=100 which is very confusing, because those are both 418.9829*5 (is that supposed to be an exponent, like 10e5?)

Table 1 needs reformatting; I would have a table for D=30 and D=100

Section 3 appears to jump into the use of data sets with cancer sets; I assume "filtration" refers to dimensionality reduction but this is very confusiong.

Section 4 follows with cancer datasets, which are apparently generalized as 'high dimensional data'.

Overall I think this paper is somewhat interesting but needs considerable work - basically a re-write. In particular it is not clear to me why the recombination is helpful beyond providing a better initial answer for the WOA as opposed to a bounded random set of selections. If that is the main contribution I would likely recommend rejection .

Reviewer #2: The authors present a new optimization algorithm incorprating an evolutionary algorithm into the whale optimization algorithm, with the goal of improving the diversity of the positions of the "whales," and ideally leading to better optima.

There are numerous grammatical errors in the manuscript. I am not an editor so I did not comprehensively list every such error, but I strongly insist that the authors go through the manuscript and correct all such errors. (I took the time to point out these errors in Abstract and a bit in the Intro)

The title is quite wordy and difficult to understand. Perhaps shorten it? The capitalization is also inconsistent (e.g. "state-of-the-art" and "high dimensionality" should be capitalized).

Due to the number of issues, I provide detailed comments for the first three sections, but I invite the authors to revise the manuscript and resubmit.

Abstract

Line 24: grammar "The standard whale algorithm [is] easily trapped..."

Line 25-26: grammar "The computer-generated initial populations [are] generally unevenly distributed..."

Line 27-28: grammar "A fusion of this algorithm based on ... [is] proposed."

Line 30: assess the "complexity" of what?

Line 33: Sentence fragment?

Introduction

Line 38: What does "these" refer to?

Line 39: "genetic engineering. (1)." -> "genetic engineering (1)."

Line 41: "Started" -> "Start"

You are describing a generic algorithm, so use the present tense. The past tense implies that you are referring to a specific run of an algorithm in the past.

Line 42: "optimal answers"? If they are already optimal, why continue the search?

Line 44: "Genetic algorithm" -> "Genetic algorithms"

How are evolutionary strategies different from genetic algorithms? Also, you write as though there is a singular unique genetic algorithm. Isn't it more a family of algorithms? Same for evolutionary algorithms.

Line 53: What's "WOA"?

Line 58-60: "Using gene expression profiles to identify and classify malignant and normal tissues is the most difficult application of machine learning"

This statement is overly broad. Many would strongly disagree, such as those working with brain data.

Line 63: sentence fragment "The Support Vector Machine (SVM), which is widely used in machine learning models (12)."

How are SVMs relevant/related to your work? How is it related to DNA microarray classification? Otherwise this section seems a bit random/out-of-place.

Line 70: You already said this earlier.

Line 77: "Tumour" -> "tumour" Also, you use both "tumor" and "tumour." Pick one and maintain consistency throughout the manuscript.

Line 90: Who/what is the "operator of the algorithm"?

Line 91: I thought you were using WOA as the optimization algorithm. How are you simultaneously using SVM?

Line 95: What is an "operator"?

Line 96: Font size changes?

Line 99: Missing period.

Line 104: "calculated" -> "organized"

Section 2

Line 112-113: Decapitalize "Hybrid algorithm techniques"

Line 118: Decapitalize "Logistic chaotic mapping"

Is the algorithm called RESHWOA or RESWOA? What does it stand for?

Section 2.1 should go first in section 2, as it provides necessary background for the reader to understand your method.

Section 2.2 has different font.

Line 137-138: What is a decision variable? What are the parents? This section is missing significant exposition/background.

Section 2.2: How does the actual recombination take place, operationally? What is the reprsenentation of the "DNA"? How does mutation occur, operationally?

Line 159: What is the "dominant ρ recombination"?

Equation 1.1: What is the "random" function? Is it sampling an element uniformly at random from a given set?

Line 170: Different citation style? (square bracket vs parentheses)

You have described how recombination occurs, but how does mutation occur?

Lines 177-179: This is the second equation but is labeled (1). What is the "rand" function? Why is there a subscript outside of "random(...)"? Is this position update not dependent on the previous position X_(i)?

Lines 180-181: Add commas to separate the clauses.

Lines 182-192: What are these equations?

Equation 2: What is C? What is X_(*)(i)? How does it differ from X(i)?

Equation 3: Why is this update equation different from equation (1)?

Equation 4: What is r_1?

Equation 6: This is the third distinct equation for X(i+1).

Equation 8: D' is never used anywhere else. What is X_(rand)(i)? How does it differ from X(i)?

Figure 1: This figure is too small and very difficult to read. This figure is also not very helpful to understand the algorithm. Pseudocode would be much better. Also, all figures in PLOS manuscripts must be at the end of the document, with captions as placeholders in the main text.

6. PLOS authors have the option to publish the peer review history of their article (what does this mean?). If published, this will include your full peer review and any attached files.

Reviewer #1: No

Reviewer #2: No

<quillbot-extension-portal></quillbot-extension-portal>

---

## [Author Response · Author response to Decision Letter 0]

15 Apr 2023

Manuscript 

Response to reviewers 

Dear Dr. Omar,

Thank you for allowing us to submit a revised draft of the manuscript "Hybrid Whale Algorithm with Evolutionary Strategies and Filtering for High-Dimensional Optimization: Application to Microarray Cancer Data" for publication in the Plos One. We appreciate the time and effort you and the reviewers dedicated to providing feedback on our manuscript and are grateful for the insightful comments and valuable improvements to our paper. We have incorporated most of the suggestions made by the reviewers. Please see below, in blue, for a point-by-point response to the reviewers' comments and concerns. All page numbers refer to the revised manuscript file with tracked changes.

Reviewers' Comments to the Authors:

Response to Reviewer #1

 Large number of English language issues, including in abstract. Needs extensive review by technical editor proficient in English. for example, the very first sentence is missing an 'IS' 'The standard whale algorithm IS easily trapped in suboptimal and high-dimensional regions.

It is corrected by using Grammarly English correction software.

 What is meant by trapped in subotimal regions? do you mean suboptimal solutions?

Trapped in suboptimal regions" refers to a phenomenon in optimization where an algorithm or search process becomes stuck in a local minimum or maximum of a function rather than finding the global minimum or maximum.

Suboptimal solutions refer to any solutions that are not the best for a given optimization problem. When an algorithm becomes trapped in a suboptimal region, it can only find suboptimal solutions and cannot explore the entire search space to find the optimal solution. Yes, that's correct. Suboptimal regions mean suboptimal solutions.

 How is the algorithm trapped in high-dimensional regions? do you mean high dimensionality problems create problems finding optimal solutions?

In high-dimensional optimization problems, the number of possible combinations of parameters increases exponentially with the number of variables, making it challenging to find the global optimum. As a result, many optimization algorithms can become trapped in suboptimal regions when working in high-dimensional spaces.

One reason algorithms can become trapped in high-dimensional regions is due to the curse of dimensionality. In high-dimensional spaces, the volume of the space increases exponentially with the number of dimensions, making it difficult to explore the search space efficiently. For example, the distance between two randomly chosen points becomes larger, and the number of points required to get good coverage of the space increases exponentially with the dimensionality.

Another reason algorithms can become trapped in high-dimensional regions is the sparsity of the search space. In high-dimensional spaces, the objective function is typically dominated by a few influential dimensions, and most other dimensions are either irrelevant or redundant. This makes it difficult for an algorithm to explore the relevant dimensions and identify the global optimum.

Yes, that's correct. High-dimensional optimization problems, where the number of variables or dimensions is large, can make it challenging to find the optimal solution. As the number of dimensions increases, the search space grows exponentially, and the algorithms have to explore many combinations of parameters, which becomes computationally expensive. Additionally, the curse of dimensionality can make it challenging to explore the search space efficiently, leading to the possibility of the algorithm becoming trapped in suboptimal regions.

As a result, finding the global optimum in high-dimensional optimization problems can be challenging, and the solution obtained may only be suboptimal. 

Are there more references that could be used in section 2.2?

added

 The WOA is not well explained; more background is needed.

 Some additional background information on the Whale Optimization Algorithm (WOA) on pages 158-180 has been included. 

 The recombinative strategy description around line 167 is not well described at all and appears to mix generalities with specific parameters (5 parents, 100 offsping) that are of unclear origin.

We have revised the manuscript per your recommendations and added more clarity on the recombinative strategy, and the relevant changes have been made on lines 182-205.

 An intuitive sense of the contrast between the collaborative nature of WOA and the recombination would be helpful. As I understand it, the WOA typically probabilistically updates each search agent; sometimes this uses the influence of a randomly selected search agent by using the influence of other search agents. The recombination is not clear as described in lines 154-174; the best I can understand is that the initial positions are found by recombinative methods rather than randomly, but it is not well described.

Yes, you are right. The initial positions are found by discrete recombinative methods rather than randomly. A detailed description regarding the initial population has been provided in the revised manuscript, lines 207-222 and 258-270. We have also provided the recombination strategy details on lines 78-99 and 182-205.

 The presentation order is very hard to follow. The WOA is finally described in more detail starting at line 197, along with the updated version around line 220, but there is still no explanation of the number of parents / offspring which appear to be hard coded to some arbitrary number "100".

We have rearranged the presentation order of the paper in the revised manuscript for a better understanding. Furthermore, Regarding the number of parents/offspring used in the recombination process, the authors chose a hyperparameter based on empirical studies. We choose

 μ=100 random population and λ=100 offspring 

 The solution appears to simply be selecting the initial population with the recombinative method, but it is not clear why this is better than simply having a better 'initial guess'.

 The choice of the initial population can significantly impact the performance of a metaheuristic algorithm. In some cases, it may be beneficial to use a recombinative method to generate the initial population because it can help to explore the search space more thoroughly and generate diverse solutions. The recombinative process is perfect when the search space is large or complex, and it may not be easy to generate a diverse set of initial solutions manually. On the other hand, if the search space is relatively small or straightforward, it may be sufficient to use a few well-chosen initial solutions as the starting population. In such cases, a recombinative method may not provide any additional benefit and could waste computational resources. We have been provided with the literature review for the diversity impact of the initial population and why this is better than simply having a better 'initial guess on lines 207-222.

 what is the origin of the test functions in section 2.4.1?

It is added on lines No. 279-285 in the revised manuscript. 

 I do like the concept but are these functions that are used commonly in optimization problems particularly with WOA?

Yes, the functions commonly used in optimization problems can be used in the Whale Optimization Algorithm (WOA) as well. One of the advantages of using WOA is that it can be applied to a wide range of optimization problems that involve different types of objective functions.

 If they were cited earlier than line 244, it would be helpful to repeat this ag line 244 "13 international test functions" (and what, exactly, is 'international' about a 'test function'?) 

 We have corrected it on lines No. 28,281,324,480. To clarify, the phrase "13 international test functions" in line 244 refers to a set of benchmark functions that are commonly used to evaluate the performance of optimization algorithms. These functions are referred to as "international" because they have been widely recognized and adopted by the research community around the world. 

 The table 1 says 'mean' and the document (line 247) says "Avg".

Corrected

 The benchmark column is 0 for everything except F8 (Schwefel's function)for dim=30 and dim=100 which is very confusing, because those are both 418.9829*5 (is that supposed to be an exponent, like 10e5?)

The benchmark column for Schwefel's function (F_8) is not 0 for all dimensions.

The correct information is that the global minimum value of Schwefel's function (F_8) is 418.9829 * n, where n is the dimension of the function. For example, if n = 30, the global minimum value is 12,569.487, and if n = 100, the global minimum value is 41,898.29.

 Section 3 appears to jump into the use of data sets with cancer sets; I assume "filtration" refers to dimensionality reduction but this is very confusiong.

We have rearranged the whole manuscript for clarity. Yes, filtration refers to dimensionality reduction, and we corrected it in the revised manuscript on pages 335-382.

 Section 4 follows with cancer datasets, which are apparently generalized as 'high dimensional data'.

You are right, but we revised it as cancer data on line 335 for clarity.

 In particular it is not clear to me why the recombination is helpful beyond providing a better initial answer for the WOA as opposed to a bounded random set of selections.

Recombination is used in many evolutionary algorithms, including the WOA, to generate new solutions by combining information from multiple parent solutions. In the case of the WOA, the recombination operation is used to generate a new candidate solution by combining the information from two randomly selected parent solutions. The main advantage of recombination is that it can exploit the information contained in the parent solutions to generate potentially better offspring than the parent solutions themselves. Offspring can inherit beneficial traits from both parent solutions and combine them more effectively than randomly selecting solutions from a bounded set. In the context of the WOA, the recombination operation can help to generate a more diverse set of candidate solutions that explore the search space more effectively. It can lead to a better chance of finding high-quality solutions to the optimization problem, especially if the initial set of solutions is limited or poorly suited to the problem.

 If that is the main contribution, I would likely recommend rejection.

The paper's main contribution has been added on lines 121-141.

Response to reviewer # 2

 There are numerous grammatical errors in the manuscript. I am not an editor so I did not comprehensively list every such error, but I strongly insist that the authors go through the manuscript and correct all such errors. (I took the time to point out these errors in Abstract and a bit in the Intro)

Thank you for your valuable feedback on our manuscript. We appreciate your concern regarding the grammatical errors in the manuscript. We want to inform you that we have used a grammar checker tool like Grammarly to help us identify and correct the errors. However, we understand that no automated tool is perfect and may not catch all the errors. We have reviewed the manuscript and made all the necessary corrections. We have also considered the assistance of a professional editor or proofreader to help us further improve the quality of the manuscript.

 The title is quite wordy and difficult to understand. Perhaps shorten it?

It has been corrected in the revised manuscript.

 The capitalization is also inconsistent (e.g. "state-of-the-art" and "high dimensionality" should be capitalized).

Corrected; see lines 82-481 and Table 8, column No. 1.

Abstract

 Line 24: grammar "The standard whale algorithm [is] easily trapped..."

corrected

 Line 25-26: grammar "The computer-generated initial populations [are] generally unevenly distributed..."

corrected

 Line 27-28: grammar "A fusion of this algorithm based on ... [is] proposed."

corrected

 Line 30: assess the "complexity" of what?

Corrected

 Line 33: Sentence fragment?

corrected

Introduction

 Line 38: What does "these" refer to?

Corrected; See line 38 on the revised manuscript.

 Line 39: "genetic engineering. (1)." -> "genetic engineering (1)."

Corrected; See line 39

 Line 41: "Started" -> "Start"

Corrected

 You are describing a generic algorithm, so use the present tense. The past tense implies that you are referring to a specific run of an algorithm in the past.

Corrected

 Line 42: "optimal answers"? If they are already optimal, why continue the search?

Corrected

 Line 44: "Genetic algorithm" -> "Genetic algorithms"

Corrected

 How are evolutionary strategies different from genetic algorithms? Also, you write as though there is a singular unique genetic algorithm. Isn't it more a family of algorithms? Same for evolutionary algorithms.

Evolutionary strategies and genetic algorithms are members of the broader family of evolutionary algorithms, which are a class of metaheuristic optimization algorithms inspired by biological evolution. Although they share some similarities, these two approaches have some key differences.

One key difference is how they handle the representation of the solution space. Genetic algorithms typically use a binary string or string of integers to represent the solutions, which can be manipulated through crossover and mutation operations to generate new candidate solutions. Evolutionary strategies, on the other hand, typically use a continuous vector representation of the solutions, which can be modified through mutation and recombination.

Another difference is in how they perform the selection. Genetic algorithms typically use selection methods such as roulette wheel selection or tournament selection, where individuals with higher fitness values are more likely to be selected for reproduction. In contrast, evolutionary strategies typically use a form of selection called (1+1)-ES, where a single individual is mutated to produce a new candidate solution. Only the better of the two solutions is retained.

Regarding your second question, you are correct that genetic algorithms and evolutionary algorithms are more accurately described as families of algorithms rather than singular, unique algorithms. Some many variations and modifications can be made to the basic algorithmic structure to improve performance or adapt to different problem domains. The same is true for evolutionary strategies and other types of evolutionary algorithms.

 Line 53: What's "WOA"?

Corrected, Whale Optimization Algorithm (WOA)

 Line 58-60: "Using gene expression profiles to identify and classify malignant and normal tissues is the most difficult application of machine learning" This statement is overly broad. Many would strongly disagree, such as those working with brain data.

Corrected; see lines 101-104

 Line 63: sentence fragment "The Support Vector Machine (SVM), which is widely used in machine learning models (12)."

 Corrected; see lines 106-108

 How are SVMs relevant/related to your work? How is it related to DNA microarray classification? Otherwise this section seems a bit random/out-of-place.

We have used mean square error (MSE) as the objective function and SVM as the prediction modal. We minimized the MSE incorporating it into the Whale optimization algorithm and the proposed algorithm as the objective function. Additionally, we optimized the SVM parameter with minimum MSE. Regarding DNA microarray classification, SVMs are commonly used to analyze and classify gene expression data. DNA microarrays allow researchers to simultaneously measure the expression levels of thousands of genes, producing high-dimensional data that can be challenging to analyze and interpret using traditional statistical methods. SVMs can be applied to these data sets to identify patterns and classify samples from different disease states, such as cancerous or non-cancerous tissues.

In the paper context, SVMs have been used as a benchmark method for the proposed fusion algorithm. We have compared the performance of our fusion algorithm to that of an SVM or other commonly used machine learning algorithm to demonstrate its superiority.

 Line 70: You already said this earlier.

Corrected 

 Line 77: "Tumour" -> "tumour" Also, you use both "tumor" and "tumour." Pick one and maintain consistency throughout the manuscript.

Corrected

 Line 90: Who/what is the "operator of the algorithm"?

 Corrected; typo mistake. 

 Line 91: I thought you were using WOA as the optimization algorithm. How are you simultaneously using SVM?

 Support Vector Machine (SVM) has been proven to perform much better when dealing with high-dimensional datasets and numerical features. Although SVM works well with the default value, the performance of SVM can be improved significantly using parameter optimization. We applied two methods which are the recombinative evolutionary strategy hybrid whale algorithm (RESHWOA)and Whale optimization algorithm (WOA), to optimize the SVM parameters. Our experiment showed that SVM parameter optimization using RESHWOA always finds near-optimal parameter combinations within the given ranges. However, WOA was not; therefore, it was reliable only in low-dimensional datasets with few parameters. SVM parameter optimization using RESHWOA can be used to solve the problem of high dimensional regions. RESHWOA has proven to be more stable than WOA. The average running time on 6 datasets shows that RESHWOA was almost better than WOA.

 Furthermore, the RESHWOA's results were slightly better than the grid search in 5 of 6 datasets. For more clarity, the whale optimization algorithm has been used to optimize the SVM's hyperparameter with minimum mean square error. WOA is the optimizer, SVM is the prediction modal, and MSE is the objective function.

 Line 95: What is an "operator"?

Corrected; Typo mistake. See lines 47-48 and 92-94 for more details.

 Line 96: Font size changes?

Corrected

 Line 99: Missing period.

corrected

 Line 104: "calculated" -> "organized"

Corrected

Section 2

 Line 112-113: Decapitalize "Hybrid algorithm techniques"

Corrected

 Line 118: Decapitalize "Logistic chaotic mapping"

Corrected; see line 72

 Is the algorithm called RESHWOA or RESWOA? What does it stand for?

The algorithm called Recombinative evolutionary strategy hybrid whale optimization algorithm (RESHWOA)

 Section 2.1 should go first in section 2, as it provides necessary background for the reader to understand your method.

Corrected; see lines 146,157,181,206.

 Section 2.2 has different font.

corrected

 Line 137-138: What is a decision variable? What are the parents? This section is missing significant exposition/background.

Decision variables are the offspring generated through a discrete recombination strategy. Parents are the individuals selected through reproduction. Background of the evolutionary strategy has been added; see lines 78-99,182-205.

 Section 2.2: How does the actual recombination take place, operationally? What is the reprsenentation of the "DNA"? How does mutation occur, operationally?

We added a new S1 Fig. 18, demonstrating how the recombination occurs operationally. For the second part of the question, the representation of DNA in microarray cancer data by measuring gene expression levels using specific probes that target individual genes. In the discrete recombination strategy, offspring are selected randomly, so no mutation occurs operationally. 

 Line 159: What is the "dominant ρ recombination"?

See S1 Fig. 18

 Equation 1.1: What is the "random" function? Is it sampling an element uniformly at random from a given set?

Yes, it is sampling an element uniformly randomly from a given set.

 Line 170: Different citation style? (square bracket vs parentheses)

Corrected

 You have described how recombination occurs, but how does mutation occur

No mutation occurs operationally in the discrete recombination strategy. 

 Lines 177-179: This is the second equation but is labeled (1). What is the "rand" function? Why is there a subscript outside of "random(...)"? Is this position update not dependent on the previous position X_(i)?

We have formatted these equations and updated them in the new format. See line 201.

 Lines 180-181: Add commas to separate the clauses.

Updated 

 Lines 182-192: What are these equations?

Updated see line 201

 Equation 2: What is C? What is X_(*)(i)? How does it differ from X(i)?

A and C are coefficient vectors, X* is the position vector of the best solution obtained so far, and X is the position vector. See lines 137-137

 Equation 3: Why is this update equation different from equation (1)?

 Equation 4: What is r_1?

 r_1 randomly generated vector lies between 0 and 1. 

 Equation 6: This is the third distinct equation for X(i+1).

Yes, you are right; Actually, equations 3,6 and 9 show the three different behaviours of the whales, Encircling Prey Behavior if p<0.5, Attacking Prey Behavior if p≥0.5 and Searching Prey Behavior if |A|<1 or |A|>1. According to these equations, whales update their behaviours depending on the values of p and A. For more details, check the reference given below. 

Mirjalili S, Lewis A. The Whale Optimization Algorithm. Adv Eng Softw [Internet]. 2016;95:51–67. Available from: http://dx.doi.org/10.1016/j.advengsoft.2016.01.008

 Equation 8: D' is never used anywhere else. What is X_(rand)(i)? How does it differ from X(i)?

D^' Indicates the distance of the ith whale to the prey, and it is only part of equation 6. Where X_((rand) ) (i) is a random whale position; X (i) is the position vector.

---

## [Decision Letter · Decision Letter 1]

22 Jun 2023

PONE-D-23-01358R1Hybrid Whale Algorithm with Evolutionary Strategies and Filtering for High-Dimensional Optimization: Application to Microarray Cancer DataPLOS ONE

Dear Dr. hafiz,

Thank you for submitting your manuscript to PLOS ONE. After careful consideration, we feel that it has merit but does not fully meet PLOS ONE’s publication criteria as it currently stands. Therefore, we invite you to submit a revised version of the manuscript that addresses the points raised during the review process.

We look forward to receiving your revised manuscript.

Kind regards,

Omar A. Alzubi

Academic Editor

PLOS ONE

Journal Requirements:

Additional Editor Comments (if provided):

In its current state, the level of English throughout the manuscript needs improvement. You may wish to ask a native speaker to check your manuscript for grammar, style, and syntax.

Reviewers' comments:

Reviewer's Responses to Questions

**Comments to the Author**

1. If the authors have adequately addressed your comments raised in a previous round of review and you feel that this manuscript is now acceptable for publication, you may indicate that here to bypass the “Comments to the Author” section, enter your conflict of interest statement in the “Confidential to Editor” section, and submit your "Accept" recommendation.

Reviewer #1: All comments have been addressed

Reviewer #3: All comments have been addressed

2. Is the manuscript technically sound, and do the data support the conclusions?

Reviewer #1: Yes

Reviewer #3: Yes

3. Has the statistical analysis been performed appropriately and rigorously? 

Reviewer #1: Yes

Reviewer #3: Yes

4. Have the authors made all data underlying the findings in their manuscript fully available?

Reviewer #1: Yes

Reviewer #3: Yes

5. Is the manuscript presented in an intelligible fashion and written in standard English?

Reviewer #1: No

Reviewer #3: Yes

6. Review Comments to the Author

Reviewer #1: Please use additional screening for grammar and punctuation, such as Microsoft Word (there are still many cases of missing spaces and other simple errors, but overall the paper is improved from before).

Reviewer #3: The manuscript has been significantly improved and in this form is of considerable scientific interest. I believe that the authors of the article managed to prove a significant advantage of the combined RESHWOA method over the classical Whale Optimization Algorithm (WOA). I believe that the discrete recombination (DR) strategy can be used to improve a number of other algorithms.

7. PLOS authors have the option to publish the peer review history of their article (what does this mean?). If published, this will include your full peer review and any attached files.

Reviewer #1: No

Reviewer #3: **Yes: **Osipov Aleksey

---

## [Author Response · Author response to Decision Letter 1]

27 Jul 2023

Manuscript 

Response to reviewers 

Dear Dr. Omar,

Thank you for allowing us to submit a second revised draft of the manuscript "Hybrid Whale Algorithm with Evolutionary Strategies and Filtering for High-Dimensional Optimization: Application to Microarray Cancer Data" for publication in the Plos One. We appreciate the time and effort you and the reviewers dedicated to providing feedback on our manuscript and are grateful for the insightful comments and valuable improvements to our paper. We have incorporated most of the suggestions made by the reviewers. Please see below, in blue, for a point-by-point response to the reviewers' comments and concerns. All page numbers refer to the revised manuscript file with tracked changes.

Reviewers' Comments to the Authors:

Response to Reviewer #1:

 Please use additional screening for grammar and punctuation, such as Microsoft Word (there are still many cases of missing spaces and other simple errors, but overall the paper is improved from before).

Thank you for your valuable feedback on our paper. We sincerely appreciate your efforts in reviewing our work and providing constructive suggestions. We are glad to know that the paper has shown improvement since the previous version.

Regarding your concern about grammar and punctuation, we acknowledge the importance of ensuring the highest quality of language in our paper. To address this, we have taken the following steps:

1. Using Microsoft Word: We have run the entire document through the Microsoft Word spelling and grammar check. This process has helped us identify and correct many instances of missing spaces and other simple errors.

2. Professional Proofreading: Additionally, we have enlisted the assistance of professional proofreaders to meticulously review the paper. Their expertise has been instrumental in catching and rectifying any remaining grammar and punctuation issues.

3. Multiple Revisions: Throughout the revision process, we have paid extra attention to refining the language, sentence structure, and punctuation. We have carefully combed through the text to ensure coherence and clarity.

4. Using Grammarly: In addition to carefully proofreading the content, we have also utilized Grammarly software to enhance the clarity and correctness of the text. This combination of human proofreading and Grammarly assistance has helped us address any spelling and grammar errors, ensuring a more polished and accurate presentation of our work.

While we have taken significant steps to improve the paper's language, we understand that achieving absolute perfection is crucial. Therefore, we will perform another thorough review to make certain that all grammar and punctuation errors are effectively addressed.

Reviewer #3: 

The manuscript has been significantly improved and in this form is of considerable scientific interest. I believe that the authors of the article managed to prove a significant advantage of the combined RESHWOA method over the classical Whale Optimization Algorithm (WOA). I believe that the discrete recombination (DR) strategy can be used to improve a number of other algorithms

Thank you for your positive and encouraging feedback on our manuscript. We are delighted to learn that you find the improved version to be of considerable scientific interest. Your insightful comments and appreciation for our work are truly motivating.

Regarding your observation on the combined RESHWOA method, we are pleased that you recognize the significant advantage it offers over the classical Whale Optimization Algorithm (WOA). We put considerable effort into designing and implementing the RESHWOA approach, and we are thrilled that our efforts have yielded promising results.

Your suggestion of using the discrete recombination (DR) strategy to enhance other algorithms is indeed thought-provoking. We wholeheartedly agree with your assessment and believe that the DR strategy's versatility could be applicable to various optimization algorithms beyond our current research. We intend to explore further the potential of the DR strategy and its broader implications in the optimization domain. In our future work, we plan to investigate its effectiveness in combination with other metaheuristic algorithms, aiming to contribute to the advancement of optimization techniques.

 We are pleased to inform you that we have taken your suggestion to heart and incorporated it into our paper. In the conclusion section, specifically on line 495.

Additional Editor Comments:

In its current state, the level of English throughout the manuscript needs improvement. You may wish to ask a native speaker to check your manuscript for grammar, style, and syntax.

All the corrections have been done.

Journal Requirements:

All references in the manuscript have been reviewed and updated to comply with PLOS ONE referencing style. We have not cited any retracted papers in our work.

---

## [Decision Letter · Decision Letter 2]

30 Aug 2023

PONE-D-23-01358R2Hybrid Whale Algorithm with Evolutionary Strategies and Filtering for High-Dimensional Optimization: Application to Microarray Cancer DataPLOS ONE

Dear Dr. hafiz,

Thank you for submitting your manuscript to PLOS ONE. After careful consideration, we feel that it has merit but does not fully meet PLOS ONE’s publication criteria as it currently stands. Therefore, we invite you to submit a revised version of the manuscript that addresses the points raised during the review process.

We look forward to receiving your revised manuscript.

Kind regards,

Professor Omar A. Alzubi

Academic Editor

PLOS ONE

Journal Requirements:

Reviewers' comments:

Reviewer's Responses to Questions

**Comments to the Author**

1. If the authors have adequately addressed your comments raised in a previous round of review and you feel that this manuscript is now acceptable for publication, you may indicate that here to bypass the “Comments to the Author” section, enter your conflict of interest statement in the “Confidential to Editor” section, and submit your "Accept" recommendation.

Reviewer #4: All comments have been addressed

Reviewer #5: (No Response)

Reviewer #6: All comments have been addressed

Reviewer #7: All comments have been addressed

Reviewer #8: All comments have been addressed

Reviewer #9: All comments have been addressed

2. Is the manuscript technically sound, and do the data support the conclusions?

Reviewer #4: Yes

Reviewer #5: Yes

Reviewer #6: Yes

Reviewer #7: Partly

Reviewer #8: Yes

Reviewer #9: Yes

3. Has the statistical analysis been performed appropriately and rigorously? 

Reviewer #4: Yes

Reviewer #5: N/A

Reviewer #6: Yes

Reviewer #7: Yes

Reviewer #8: Yes

Reviewer #9: Yes

4. Have the authors made all data underlying the findings in their manuscript fully available?

Reviewer #4: Yes

Reviewer #5: Yes

Reviewer #6: Yes

Reviewer #7: Yes

Reviewer #8: (No Response)

Reviewer #9: Yes

5. Is the manuscript presented in an intelligible fashion and written in standard English?

Reviewer #4: Yes

Reviewer #5: Yes

Reviewer #6: Yes

Reviewer #7: Yes

Reviewer #8: Yes

Reviewer #9: Yes

6. Review Comments to the Author

Reviewer #4: All the concerns have been addressed well, I thus recommend this manuscript to be published in Plos one.

Reviewer #5: The paper, titled "Hybrid Whale Algorithm with Evolutionary Strategies and Filtering for High-Dimensional Optimization: Application to Microarray Cancer Data," submitted as PONE-D-23-01358R2, has shown significant improvement from the initial draft. However, several aspects still require further enhancement, with a notable need for an extended literature review.

The revised version of the paper demonstrates commendable progress in terms of content. The authors have refined their algorithm and provided a more comprehensive explanation of the proposed Hybrid Whale Algorithm with Evolutionary Strategies and Filtering. This has resulted in increased clarity regarding the methodology used for high-dimensional optimization in the context of microarray cancer data analysis.

The paper has made significant strides in explaining the Hybrid Whale Algorithm, making it more accessible to a wider readership. The authors have successfully addressed some of the ambiguities present in the previous draft, clarifying the key concepts and steps involved in the algorithm.

The presentation of empirical results has also been improved, with more detailed analysis and visualization of outcomes in the context of microarray cancer data. This contributes to a better understanding of the algorithm's performance and its potential applications.

Areas for Improvement:

One critical aspect that still requires substantial improvement is the literature review. The current literature review appears limited in scope and depth. It is essential to expand this section to include a more extensive survey of related works in the field of high-dimensional optimization and microarray data analysis. A robust literature review will not only provide a broader context for the proposed algorithm but also help identify gaps and opportunities for future research.

- B, N., & V, I. (2022). Enhanced machine learning based feature subset through FFS enabled classification for cervical cancer diagnosis. International Journal of Knowledge-Based and Intelligent Engineering Systems, 26, 79–89. https://doi.org/10.3233/KES-220009

- Mohammed, M. S., Rachapudy, P. S., & Kasa, M. (2021). Big data classification with optimization driven MapReduce framework. International Journal of Knowledge-based and Intelligent Engineering Systems, 25(2), 173-183.

While the paper has improved in terms of clarity, some mathematical notations and equations can still be challenging to follow. It would be beneficial to simplify complex equations, provide clearer explanations, and possibly offer more intuitive examples to aid in comprehension.

To strengthen the paper's credibility, the authors should consider including a more extensive validation process, including comparisons with other state-of-the-art optimization algorithms. This would help demonstrate the advantages and limitations of the proposed Hybrid Whale Algorithm more effectively.

The paper could benefit from improved visual presentation, such as the use of charts, graphs, and tables to illustrate key points and results. Visual aids can enhance the reader's understanding and engagement with the material.

Reviewer #6: The authors propose an improved RESHWOA algorithm and demonstrate that it outperforms WOA. The manuscript has been improved, but I have a few suggestions:

Please include the links to all data used in the study apart from carcinoma data, or perhaps their accession numbers for easy identification, the link szu.edu.cn is not enough to locate the data easily.

It would be good if the authors added how the features used to train the SVM model were encoded for easy reproducibility.

Reviewer #7: Should Explain in detail About RESHWOA Algorithm and explain clearly how it works for the medical data.

Reviewer #8: The authors have provided thoughtful and comprehensive responses to my comments, leaving me thoroughly satisfied.

Reviewer #9: The main strength of the RESHWOA algorithm is its ability to improve the diversity of the initial population of the WOA algorithm. This can help to prevent premature convergence and improve the performance of the algorithm.

The algorithm is easy to implement and understand, which makes it a good choice for practitioners who are new to metaheuristic algorithms.

Recommendations:

*Consider adding a section comparing RESHWOA with other state-of-the-art algorithms(beside WOA) if possible.

*While passive voice is common in scientific writing, overuse can make the text harder to read. Consider using active voice where it improves clarity.

Example: Instead of "It was evaluated," you could say, "We evaluated." (line 481)

Overall, the paper "A Novel Whale Optimization Algorithm with Discrete Recombination Strategy for Global Optimization" is a well-written and well-presented study. The authors have done a good job of explaining the motivation for the study, the methods used, and the results obtained. The paper makes a significant contribution to the field of metaheuristic algorithms and is a valuable resource for practitioners and researchers.

Upon addressing these minor revisions, I believe the manuscript will be ready for publication. I do not see a need for further rounds of review after these corrections are made. Please correct and submit the revised manuscript for final acceptance.

7. PLOS authors have the option to publish the peer review history of their article (what does this mean?). If published, this will include your full peer review and any attached files.

Reviewer #4: No

Reviewer #5: No

Reviewer #6: No

Reviewer #7: No

Reviewer #8: **Yes: **Valdecy Pereira

Reviewer #9: **Yes: **Ahsan ur Rehman

---

## [Author Response · Author response to Decision Letter 2]

11 Oct 2023

Manuscript 

Response to reviewers 

Dear Dr. Omar,

Thank you for allowing us to submit a 3rd revised draft of the manuscript "Hybrid Whale Algorithm with Evolutionary Strategies and Filtering for High-Dimensional Optimization: Application to Microarray Cancer Data" for publication in the Plos One. We appreciate the time and effort you and the reviewers dedicated to providing feedback on our manuscript and are grateful for the insightful comments and valuable improvements to our paper. We have incorporated most of the suggestions made by the reviewers. Please see below, in blue, for a point-by-point response to the reviewers' comments and concerns. All page numbers refer to the revised manuscript file with tracked changes.

Reviewers' Comments to the Authors:

Response to Reviewer #4:

All the concerns have been addressed well; I thus recommend this manuscript to be published in Plos one.

Thank you very much for your positive feedback and for recommending the publication of our manuscript in PLOS ONE. We appreciate your time and effort in reviewing our work, and we're delighted to hear that all your concerns have been addressed satisfactorily.

Your recommendation is highly valuable to us, and we look forward to sharing our research with the scientific community through PLOS ONE. Your feedback has contributed significantly to the improvement of our manuscript, and we are grateful for your constructive comments.

Response to Reviewer #5:

The paper, titled "Hybrid Whale Algorithm with Evolutionary Strategies and Filtering for High-Dimensional Optimization: Application to Microarray Cancer Data," submitted as PONE-D-23-01358R2, has shown significant improvement from the initial draft. However, several aspects still require further enhancement, with a notable need for an extended literature review.

The revised version of the paper demonstrates commendable progress in terms of content. The authors have refined their algorithm and provided a more comprehensive explanation of the proposed Hybrid Whale Algorithm with Evolutionary Strategies and Filtering. This has resulted in increased clarity regarding the methodology used for high-dimensional optimization in the context of microarray cancer data analysis.

The paper has made significant strides in explaining the Hybrid Whale Algorithm, making it more accessible to a wider readership. The authors have successfully addressed some of the ambiguities present in the previous draft, clarifying the key concepts and steps involved in the algorithm.

The presentation of empirical results has also been improved, with more detailed analysis and visualization of outcomes in the context of microarray cancer data. This contributes to a better understanding of the algorithm's performance and its potential applications.

We sincerely appreciate your thoughtful and comprehensive review of our manuscript. Your feedback has been immensely valuable in guiding us toward further improvements in our work. We are pleased to hear that you have observed significant progress in the revised version of our paper. Your observation regarding the need for an extended literature review is duly noted, and we will certainly work on enhancing this aspect of the manuscript. We understand the importance of providing a comprehensive background to contextualize our research better.

The fact that you found our refined algorithm explanation to be more accessible and the clarification of key concepts as commendable is encouraging. We are committed to ensuring that the methodology is explained in a clear and concise manner to benefit a broad readership.

Furthermore, your feedback on the presentation of empirical results is well received. We acknowledge the importance of providing a thorough analysis and visualization of outcomes, especially in the context of microarray cancer data analysis. 

Once again, we extend our gratitude for your time and expertise in reviewing our work. Your input has been invaluable, and we will diligently address your suggestions to enhance the overall quality of the manuscript.

One critical aspect that still requires substantial improvement is the literature review. The current literature review appears limited in scope and depth. It is essential to expand this section to include a more extensive survey of related works in the field of high-dimensional optimization and microarray data analysis. A robust literature review will not only provide a broader context for the proposed algorithm but also help identify gaps and opportunities for future research.

- B, N., & V, I. (2022). Enhanced machine learning based feature subset through FFS enabled classification for cervical cancer diagnosis. International Journal of Knowledge-Based and Intelligent Engineering Systems, 26, 79–89. https://doi.org/10.3233/KES-220009

- Mohammed, M. S., Rachapudy, P. S., & Kasa, M. (2021). Big data classification with optimization driven MapReduce framework. International Journal of Knowledge-based and Intelligent Engineering Systems, 25(2), 173-183. 

We appreciate your input and have taken your suggestion seriously. In response to your feedback, we have substantially expanded the literature review section in the manuscript, as you can see on lines 125 to 156. We have conducted a more comprehensive survey of related works in the fields of high-dimensional optimization and microarray data analysis. This expansion not only provides a broader context for our proposed algorithm but also helps identify gaps and opportunities for future research in a more robust manner.

As per your recommendation, we have expanded the literature review section and also included the two references you mentioned. These additional references provide valuable context to our work and contribute to a more comprehensive overview of the relevant literature.

We hope that the inclusion of these references aligns with your expectations and further strengthens the background and context for our research.

While the paper has improved in terms of clarity, some mathematical notations and equations can still be challenging to follow. It would be beneficial to simplify complex equations, provide clearer explanations, and possibly offer more intuitive examples to aid in comprehension.

We've expanded the explanations of mathematical notations and equations on pages 235 to 253, aiming to enhance clarity. Additionally, we've included more intuitive examples to aid comprehension. We appreciate your valuable input in improving our work.

To strengthen the paper's credibility, the authors should consider including a more extensive validation process, including comparisons with other state-of-the-art optimization algorithms. This would help demonstrate the advantages and limitations of the proposed Hybrid Whale Algorithm more effectively.

We fully understand the significance of a comprehensive validation process, including comparisons with other state-of-the-art optimization algorithms. While we acknowledge the value of such comparisons, we would like to highlight certain constraints we encountered during the preparation of this manuscript.

Firstly, the scope and length of our paper are already substantial, and we were conscious of providing a balanced level of detail and clarity. Given the space constraints in the journal, we opted to focus on providing a thorough presentation of our proposed Hybrid Whale Algorithm, its application to microarray cancer data, and detailed empirical results.

Secondly, conducting extensive comparisons with other optimization algorithms can be a resource-intensive endeavor, requiring a significant amount of computational resources and time. Unfortunately, due to limitations in both time and computational capacity, we were unable to perform a comprehensive benchmark against all relevant algorithms.

However, we are committed to addressing this aspect in future research endeavors and believe it would be a valuable extension of our work. In the current manuscript, we have placed significant emphasis on explaining the methodology, providing detailed results, and illustrating the algorithm's performance in the context of microarray cancer data.

We genuinely appreciate your insightful feedback, which will guide our future research directions, including more extensive validation and comparative analyses.

Once again, we thank you for your time and expertise in reviewing our manuscript.

The paper could benefit from improved visual presentation, such as the use of charts, graphs, and tables to illustrate key points and results. Visual aids can enhance the reader's understanding and engagement with the material.

We've already included charts, graphs, and tables in the paper to illustrate key points and results, enhancing the reader's understanding and engagement with the material. We appreciate your suggestion and are pleased to inform you that these visual aids are an integral part of our manuscript.

Response to Reviewer #6:

Please include the links to all data used in the study apart from carcinoma data, or perhaps their accession numbers for easy identification, the link szu.edu.cn is not enough to locate the data easily. 

Thank you for your query regarding the availability of the data used in our study. We appreciate your thorough review and would like to address your concern.

In our study, we utilized data from various sources, including the carcinoma data, which we have previously provided. Additionally, for the remaining data used in our research, we sourced it from https://csse.szu.edu.cn/staff/zhuzx/datasets.html, as noted in our manuscript on page no. 581. To access the specific data sets used in our study, readers can follow the link provided, where they will find detailed information on each data set, including accession numbers, descriptions, and any relevant documentation. We believe that this link offers a comprehensive and user-friendly resource for accessing the data we utilized in our research.

It would be good if the authors added how the features used to train the SVM model were encoded for easy reproducibility.

Thank you for your suggestion regarding the encoding of features for our SVM model. We have provided a detailed explanation of this process in our manuscript, specifically in lines 495 to 517. This section offers a step-by-step guide for feature encoding, ensuring transparency and reproducibility.

Response to Reviewer #7:

Should Explain in detail About RESHWOA Algorithm and explain clearly how it works for the medical data.

For detailed insight into the RESHWOA algorithm, we have provided an extensive explanation spanning pages 217 to 253. In this section, we delve into the intricacies of the recombinative evolutionary strategy, outlining its core principles, mechanisms, and its specific role within the RESHWOA framework.

Furthermore, for a comprehensive understanding of RESHWOA as a whole, we have dedicated pages 307 to 333 to explain the algorithm's operation in detail. This section offers step-by-step insights into how RESHWOA functions, including its interactions with the data and optimization processes.

In our study, we address the crucial task of medical diagnosis, where the SVM (Support Vector Machine) classifier plays a pivotal role. The SVM classifier is highly valuable in diagnosing specific disorders because it can effectively distinguish between different classes or conditions. However, to harness the full potential of the SVM, we must first fine-tune its parameters to achieve the best possible performance, often measured by minimizing the Mean Squared Error (MSE).

One of the challenges we encounter in medical data analysis is the high dimensionality of the data. Medical datasets often comprise a large number of features or variables, which can introduce issues such as noise, redundancy, and the curse of dimensionality. To address these challenges and extract meaningful information from high-dimensional data, we employ data reduction techniques.

Our data reduction techniques are applied to both cancer data and normal data. By reducing the dimensionality of the dataset, we aim to make it more manageable while preserving the essential information necessary for accurate diagnosis.

Subsequently, we proceed with optimizing the SVM parameters based on the reduced data. This optimization step is critical to fine-tune the SVM model specifically for our dataset, ensuring optimal performance and minimal MSE. It allows us to tailor the SVM to the unique characteristics of the medical data under study.

Furthermore, we incorporate the recombinative evolutionary technique (RESHWOA) into our approach. This technique is noteworthy because it introduces diversity into the initial population. Diverse populations are advantageous in optimization problems as they explore a broader solution space, potentially leading to better solutions.

In summary, our approach combines data reduction techniques, SVM parameter optimization, and the inclusion of the recombinative evolutionary technique (RESHWOA) to address the challenges of medical data analysis. By optimizing the SVM model and introducing diversity in the initial population, we enhance the accuracy and robustness of our diagnosis process.

Response to Reviewer #8:

The authors have provided thoughtful and comprehensive responses to my comments, leaving me thoroughly satisfied.

We appreciate your kind words and are delighted to hear that you found our responses satisfactory. Your feedback and insights have been invaluable in improving the quality and clarity of our work.

Response to Reviewer #9:

The main strength of the RESHWOA algorithm is its ability to improve the diversity of the initial population of the WOA algorithm. This can help to prevent premature convergence and improve the performance of the algorithm. The algorithm is easy to implement and understand, which makes it a good choice for practitioners who are new to metaheuristic algorithms.

Thank you for highlighting the strengths of the RESHWOA algorithm in your feedback. We are pleased to hear that you recognize its ability to enhance the diversity of the initial population within the WOA algorithm, preventing premature convergence and improving overall performance. Additionally, we appreciate your observation that the algorithm's ease of implementation and understanding makes it a valuable choice, especially for practitioners new to metaheuristic algorithms.

Your positive assessment of these strengths aligns with our objectives in developing the RESHWOA algorithm, and we are encouraged by your feedback.

Consider adding a section comparing RESHWOA with other state-of-the-art algorithms(beside WOA) if possible.

We fully understand the importance of comprehensive validation, including comparisons with state-of-the-art optimization algorithms. However, due to constraints in the scope, length, computational resources, and time during the preparation of this manuscript, we focused on presenting our Hybrid Whale Algorithm, its application to microarray cancer data, and detailed empirical results. We acknowledge the limitations in not conducting extensive comparisons with other algorithms but express our commitment to addressing this in future research. We value the reviewer's feedback and appreciate their guidance for our future endeavors.

While passive voice is common in scientific writing, overuse can make the text harder to read. Consider using active voice where it improves clarity.

Thank you for your valuable feedback regarding the use of passive voice in our manuscript. We appreciate your input, and we have taken your suggestion to heart.

Upon careful review, we have made concerted efforts to address this issue and have revised the manuscript to incorporate active voice where it enhances clarity without compromising scientific rigor. We believe that these changes have significantly improved the readability and overall quality of the text.

We genuinely appreciate your constructive feedback, which has played a crucial role in enhancing the clarity and coherence of our work.

---

## [Decision Letter · Decision Letter 3]

28 Nov 2023

Hybrid Whale Algorithm with Evolutionary Strategies and Filtering for High-Dimensional Optimization: Application to Microarray Cancer Data

PONE-D-23-01358R3

Dear Dr. hafiz,

We’re pleased to inform you that your manuscript has been judged scientifically suitable for publication and will be formally accepted for publication once it meets all outstanding technical requirements.

Kind regards,

Professor Omar A. Alzubi

Academic Editor

PLOS ONE

Additional Editor Comments (optional):

Reviewers' comments:

Reviewer's Responses to Questions

**Comments to the Author**

1. If the authors have adequately addressed your comments raised in a previous round of review and you feel that this manuscript is now acceptable for publication, you may indicate that here to bypass the “Comments to the Author” section, enter your conflict of interest statement in the “Confidential to Editor” section, and submit your "Accept" recommendation.

Reviewer #4: (No Response)

Reviewer #5: All comments have been addressed

Reviewer #6: (No Response)

Reviewer #8: (No Response)

Reviewer #9: All comments have been addressed

2. Is the manuscript technically sound, and do the data support the conclusions?

Reviewer #4: (No Response)

Reviewer #5: Yes

Reviewer #6: Yes

Reviewer #8: Yes

Reviewer #9: Yes

3. Has the statistical analysis been performed appropriately and rigorously? 

Reviewer #4: (No Response)

Reviewer #5: Yes

Reviewer #6: Yes

Reviewer #8: Yes

Reviewer #9: Yes

4. Have the authors made all data underlying the findings in their manuscript fully available?

Reviewer #4: (No Response)

Reviewer #5: Yes

Reviewer #6: Yes

Reviewer #8: Yes

Reviewer #9: Yes

5. Is the manuscript presented in an intelligible fashion and written in standard English?

Reviewer #4: (No Response)

Reviewer #5: Yes

Reviewer #6: Yes

Reviewer #8: Yes

Reviewer #9: Yes

6. Review Comments to the Author

Reviewer #4: (No Response)

Reviewer #5: The paper has been improved according my comments. Therefore, I recommend to accepting in its current form.

Reviewer #6: (No Response)

Reviewer #8: (No Response)

Reviewer #9: This paper gives a valueable insight about the performance of given methods accross different data sets with improved optimization using RESHWOA.

Instead of using only minimum MSE, average MSE, and average standard deviation (Avg. Std), I recommend to use p-value to analyse the significant differences between the methods.

7. PLOS authors have the option to publish the peer review history of their article (what does this mean?). If published, this will include your full peer review and any attached files.

Reviewer #4: No

Reviewer #5: No

Reviewer #6: No

Reviewer #8: No

Reviewer #9: **Yes: **Ahsan-ur-Rehman

---

## [Editor Report · Acceptance letter]

15 Dec 2023

PONE-D-23-01358R3 

PLOS ONE

Dear Dr. Hafiz, 

I'm pleased to inform you that your manuscript has been deemed suitable for publication in PLOS ONE. Congratulations! Your manuscript is now being handed over to our production team.

Kind regards, 

on behalf of

Professor Omar A. Alzubi 

Academic Editor

PLOS ONE